

# Assessment of lumped hydrological balance models in peninsular Spain

Julio Pérez-Sánchez[1], Javier Senent-Aparicio[1], Francisco Segura-Méndez[1], David Pulido-Velazquez[1]

[1]Department of Polytechnic Science, University of Polytechnic Science, UCAM University of San Antonio of Murcia, Campus
los Jerónimos, nº 135, 30107 Guadalupe. Murcia (Spain)

*Correspondence to*: Julio Pérez-Sánchez (jperez058@ucam.edu)

**Abstract.** The assessment of inflows in a water resources system is essential for the appropriate analysis of its management. These inflows can be obtained from hydrological balance models. In this paper, we intend to perform a comparative study of six lumped hydrological balance models in several basins with different climatic conditions within Spain. Lumped models enable the estimation of catchment resources without using spatially distributed information that would not be available in many cases. We have selected basins where long time series of climatic and hydrological data are available (more than 30 years) to calibrate parsimonious models, taking into account the stochastic behaviour of the natural streamflow and the climatic variables. The study period comprises 34 years (1977–2010). The explored models are Témez, ABCD, GR2M, the Australian water balance model (AWBM), GUO-5 parameters (Guo-5p) and Thornthwaite-Mather. Six statistical indices are applied to compare the results of the models: Nash–Sutcliffe model efficiency coefficient (NSE), root-mean-square deviation (RMSE), Pearson's correlation coefficient (R), percent bias (PBIAS), RMSE-observations standard deviation ratio (RSR) and the relative error between observed and simulated runoff volumes (REV). The results show that although lumped models can be employed in humid and sub-humid regions, the more humid the catchments are, the better the results obtained. Témez models provide the worst results in dry sub-humid and semi-arid regions. Guo-5p estimates runoff volumes with errors below 10% despite the unsatisfactory results provided according to the Bressiani classification. The Bressiani classification takes into account different comparison criteria to help in the decision-making process when selecting a model. Nevertheless, the assessment of the margin of error in total runoff volume using REV is also a key index. The usefulness of Pearson's correlation when selecting a model is quite low but can be helpful in the analysis of models' weaknesses.

## 1 Introduction

Water resources assessment is key to the analysis of catchment management (Wurbs, 2005). The development of models to study water-management policies is a complex task that presents a fundamental scientific challenge (Welsh, 2007). Especially complex cases occur in arid and semi-arid regions, where precipitation is limited and/or irregular and evapotranspiration (ET) rates are high. Hydrological balance models are used to reconstruct historical series and predict future ones (Puricelli, 2003). They are based on the principle of mass conservation or the continuity equation (Essam, 2007; Rose, 2004), which considers that the difference of inputs and outputs will be reflected in water storage in the catchment (Shimon, 2010; UNESCO, 1981).



The concept of hydrological balance models was first introduced by Thornthwaite (1948) and Thornthwaite and Mather (1957). They proposed two different conceptual models based on two parameters: soil moisture capacity and water excess above the maximum soil moisture storage capacity. These models demonstrated a good fit to estimate monthly runoff (Alley, 1984) and have formed the basis of many other two-parameter hydrological models (Xiong and Guo, 1999; Makhlouf and Michel, 1994;

Giakoumakis et al., 1991; Mimikou et al., 1991; Alley, 1985, 1984). There are also water balance models that comprise more than two parameters (Abulohom et al., 2001; Yates and Strzepek, 1998; Vandewiele et al., 1992). However, Xiong and Guo (1999) showed that their proposed two-parameter model in China performed as well as a five-parameter model. To date, several studies have shown that many models produce similar results to previous ones (Andreassian et al., 2006; Perrin et al., 2001; Chiew et al., 1993; World Meteorological Organization, 1975). In a lumped water balance model, catchment parameters and

variables are averaged in space, so hydrological processes are approached by conceptual solutions formulated by using semi-empirical equations. The system is described using different reservoirs, the moisture content of which depends on the relationships (physical and empirical) between them (Xu and Singh, 1998). A lumped hydrological balance model may have only three or four parameters (Xu and Singh, 1998; Vandewiele et al., 1992; Alley, 1984) and can be implemented with several lines of computer code, whereas a complex model may have more than 20 parameters (Chiew, 2010). Some examples of

lumped models are the ABCD model (Zhao et al., 2016; Wang and Tang, 2014; Sankarasubramanian and Vogel, 2002; Alley, 1985) GR2M (Lacombe et al., 2016; Mouelhi et al., 2006), Sacramento (Burnash et al., 1973), Guo-5p (Xiong and Guo, 1999; Guo, 1995), Témez (Singh and Kumar, 2016; Singh, 2000; Ferrer, 1993; Témez, 1991, 1987), Thornwaite-Mather (Lyon et al., 2004; Frankenberger et al., 1999; Calvo, 1986), IHACRES (Croke et al., 2006), SIMHYD (Chiew et al., 2002), GR4J (Perrin et al., 2003), AWBM (Boughton, 2009, 2007, 2006, 2004; Boughton and Chiew, 2007) and SMAR (O'Connel et al.,

1970). More examples of rainfall-runoff models can be found in Singh (1995) and Singh and Frevert (2002).

With the new development of computer aided tools and more detailed information, there is an increasing trend to use distributed or semi-distributed models (Eder et al., 2005; Arnold et al., 1998). They provide more detailed distributed results on a catchment scale approximating heterogeneities of the system. However, uncertainty at high resolution may diminish potential

gains in prediction accuracy (Carpenter, 2006). Nevertheless, despite the simplicity of lumped models, they perform well in many studies (Yang and Michel, 2000; Cameron et al., 1999; Uhlenbrook et al., 1999; Yang et al., 1995). However, they do not need as much data as the distributed models (which are unavailable in many cases), and the complexity and requirements to process them are lower. Furthermore, calibration of the lumped parameter models is much less time consuming and produced higher overall model performance in comparison to the more complex distributed models (Vansteenkiste et al., 2014). They

are particularly useful in small data-rich catchments and are used in conjunction with field studies (Chiew, 2010). Various studies have been conducted to compare distributed and lumped models (Koren et al., 2004; Zhang et al., 2004; Boyle et al., 2001; Refsgaard and Knudsen, 1996; Shah et al., 1996). The suitability of a model depends on the basin and specific regional characteristics.



The aim of this study is to analyse and compare six lumped water balance models in 16 basins located in different climatic regions of Spain. We have selected basins where long time series of climatic and natural streamflow data are available (more than 30 years). We intend to calibrate parsimonious models by considering the stochastic behaviour of the natural streamflow and climatic variables. The aim is to select the models with the best fit and assess the comparison methods used according to

the characteristics of the region. The contents of the paper are structured as follows: the data and study area are introduced in Section 2; methodology is described in Section 3; Section 4 presents the results and discussion and Section 5 highlights the main conclusions.

**2 Study area and data**

Spain is the second largest country in Western Europe, with a territory covering 505,990 km$^2$, a dense network of rivers with

many branches and a large number of aquifers. This disparity within the Iberian Peninsula is optimal for hydrological research. Spain features a wide range of climates due to its position between the European temperate zone and the subtropical zone. It also includes some of the rainiest areas in Europe in the northeast and the driest areas in the southeast, with a marked summer drought. To ensure the validity of the results, the 16 selected catchments are in natural regime. They are located all over the country, as shown in Figs. 1 and 2. Thus, geographic and climatic variety in Spain is reflected wherever data were available.

Their altitudes vary from 1,632 to 342 metres above sea level (MASL), and catchment areas range from 29 km$^2$ to 837 km$^2$ with an average of 300 km$^2$ (Table 1). There is no south-western catchment in this study due to the lack of data in this area, where most gauging stations have data for a period of less than 10 years. As shown in Table 1, the average temperatures range from 8–16 ºC, depending on the latitude and average altitude of the catchment, with a positive gradient to the south. With regard to the rainfall regime, the highest yearly precipitation occurs in the north of Spain, where average temperatures are

lower and, consequently, ET is less. However, in the southern half of the Iberian Peninsula, ET is generally higher than precipitation, especially in the lowlands, which have an average altitude of less than 600 metres.

The study area comprises the most common climate groups in the Iberian peninsula, according to Köppen's (1936, 1918, 1884) classification: Bsk., Csb., Csa., Cbf. and UNEP aridity index (UNEP, 1997). This index (AIU) is defined by Eq. 1.

$$AI_U = \frac{P}{PET} \tag{1}$$

where P is the average annual precipitation and PET is the potential ET. The thresholds that define the various degrees of aridity depend on the value of AIU according to Table 2.

Among the 16 catchments studied, nine are considered humid; thus, their AIU exceeds one, and five of them even have an aridity index near to or above two. All are located in northern Spain. Despite the rainfall gradient from Northwest to Southeast, the SEG and ZUM catchments are classified as humid sub-humid due to their altitude above 1,000 MASL. The BOL catchment,

which is close to the Mediterranean Sea, is the only eastern dry sub-humid region of the regions studied; thus, the other dry sub-humid catchments (TAM and CUE) are located in the centre of the Iberian Peninsula. The only semi-arid region in the studied catchments is RVA, the average altitude of which is approximately 600 MASL.





Precipitation and ET data series in each basin are from a 33-year period (1977–2010). They were obtained from the official monthly series provided by the CEDEX (Centre of Studies and Experimentation of Civil Works) for the Spanish government (Álvarez et al., 2005) at a spatial resolution of 500x500 m$^2$ (Estrela and Quintas, 1996). The model has been validated at more than 100 control points (Estrela et al., 1999) and used in Spain for water resources assessment, in the White Paper Book of

Waters (Ministry of Environment 2000) and the National Water Master Plan (Ministry of Environment 2002), and in several studies (Pérez-Martín et al., 2014; González-Zeas et al., 2012; Belmar et al., 2011). Natural streamflow data in each catchment are available for the same period. They come from measurements at gauging stations in the official Spanish network. The range of missing values moves from 2% to 8% in the stations considered.

### 3 Methodology

The methods used in this investigation are based on the identification of the hydrological models that best fit each catchment considered. Six monthly water balance models were used: Témez, ABCD, GR2M-1994, Australian Water Balance Model (AVBM), Guo-5p and Thornwaite-Mather. All these models use precipitation and potential ET as input data. Finally, the assessment of the goodness of fit for the six models was mainly performed using the ratings Moriasi et al. (2007) suggested, and the grading system proposed by Bressiani et al. (2015). We also analysed the Pearson's correlation coefficient and the

relative error between observed and simulated runoff volumes in the studied period. All of these hydrological models are defined with four parameters, except Thornwaite-Mather (three parameters) and Guo-5p. (five parameters). The models conduct different moisture balances according to the different processes in a hydrological system through all phases of the hydrological cycle. The processes are governed by the continuity principle and mass balance and remain regulated by the specific laws of division and transfer between the balance parameters (Cabezas et al., 1999).

### 3.1 Water balance models

### 3.1.1 Témez model

Témez (1977) developed the Témez model, which has been widely used in Spanish catchments (Escriva-Bou et al., 2017, MIMAM, 2000; Cabezas et al., 1999; Estrela et al., 1999) and by the Spanish government in water management (Estrela, 1992). This model considers the system to be divided into two (Fig. 3): the upper or non-saturated zone (S) and the lower or

saturated zone (G).  Some of the precipitation (P) drains directly into the river or through the aquifer, while the remainder is converted into ET. Excess is divided into runoff (Qs) through river networks at the present time, and infiltration to aquifers, draining one part (Qg) at the present time, with the rest remaining in the groundwater storage tank (G) for draining at a later date.



### 3.1.2. ABCD Model.

The ABCD 4-parameter model (Alley, 1984) introduces a different formulation in ET process and allows for a water surplus even though the soil moisture tank (S) is not full yet. As shown in Fig. 3, this model also considers two storage tanks: upper storage (S) or soil and groundwater storage (G). The upper storage tank (S) has two outputs: runoff (Qs) and infiltration. Thus, the model has two inputs, precipitation (P) and ET, and their outputs are soil moisture content at the end of the month (S), monthly available water, real ET, runoff (Qs), infiltration, groundwater runoff (Qb), monthly groundwater storage (G) and total runoff (Q) (Fig. 3).

### 3.1.3. GR2M-1994 Model (GR4-1994).

This model (Makhlouf and Michel, 1994) was developed in the 1990s by the CEMAGREF (Centre of Agricultural and Environmental Research of France). It is also based on monthly precipitation and ET (Edijatno and Michel, 1989). Afterwards, the model evolved into different versions, such as GR1A, GR2M, GR3J and GR4J, denominated the number of required parameters and the last letter denominating the period considered: J (daily), M (monthly) or A (yearly). GR2M transforms precipitation into runoff through the implementation of two equations: production and transfer functions (Kabouya, 1990; Edijatno and Michel, 1989). Initially, P and ET are balanced and precipitation is distributed between the upper storage tank (S) with a limited capacity and groundwater storage tank (G) (Paturel et al., 1995). Nief et al. (2003) showed that model parameters are robust to non-stationary rainfall series, and calibrated parameter values are highly correlated with land use. Like previous models, monthly precipitation and evapotransporation are the inputs, and the operating diagram is shown in Fig. 3.

### 3.1.4. AWBM Model.

The AWBM was also developed in the 1990s and is the most commonly used water balance method in Australia (Boughton, 2004). This model has three surface water-storage tanks (S1, S2 and S3). The water balance of each is estimated independently, resulting in three surpluses. One part of these surpluses is transformed into runoff (Qs), and the other part percolates to a groundwater storage tank or aquifer (G), which in turn goes to groundwater runoff (Qg). Total flow (Q) is obtained by adding both runoffs (Fig. 3).

### 3.1.5. Guo Model (five parameters).

This model was developed to estimate the runoff in 70 catchments in southern China. It has a similar performance to the two-parameter Guo model, and its use is particularly recommended in humid and semi-humid regions (Xiong and Guo, 1999; Guo, 1995). Precipitation and evapotranspiration (P and ET) are the input data, on the basis of which the remaining parameters are estimated: real ET, soil water storage (S), water surpluses, surface runoff (Qs), subsurface runoff (Qb), aquifer recharge, groundwater storage (G), groundwater runoff (Qg) and total flow (Q).



### 3.1.6. Thornthwaite-Mather Model.

This model was developed by Thornwaite and Mather (Thornthwaite et al., 1957, 1955) in the early 1940s for the Delaware River, and many water balance models are based on it. The model distinguishes two water storage tanks: surface (S) and groundwater (G), which lead to the output flow (Q) through different calculations.

### 3.2. Goodness-of-fit tests

The study period is 34 years (1977–2010). We employed three years for warming up, and the other 30 years, starting October 1980, for calibration and validation, applying the split sample test proposed by KlemeŠ (1986). Therefore, the data series from 1980 to 2010 (30 years) was divided into two sets. The first 15 years (1980–1995) were used for calibration, and the remaining 15 years (1995–2010) were used in model verification, namely as a security measure of calibration. Thus, model parameters that best fit the simulated data versus the observed data are obtained. The gradient reduced algorithm (GRG2) (Ríos, 1988; Abadie, 1978; Lasdon and Waren, 1978; Lasdon et al., 1978) was used to calibrate the parameters. Once the model was calibrated and the parameters validated, the predictive stage was executed.

To evaluate model accuracy, six statistic indices were obtained: NSE (Nash and Sutcliffe, 1970), RMSE (Hyndman et al., 2006), R (Pearson, 1895), PBIAS (Gupta et al., 1999), RSR (Moriasi et al., 2007) and REV (Karpouzos et al., 2011). The formulas used are presented in Table 3, with optimal values as recommended by Moriasi et al. (2007).

According to Moriasi et al. (2007) and Bressiani et al. (2015), a grading method was established to evaluate the good performance of the model based on the NSE, RSR and PBIAS values. A model was considered unsatisfactory if one of the previous tests is set as unsatisfactory according to Moriasi et al. (2007). In any other case, the grading system proposed by Bressiani et al. (2015) was applied to classify the models in four categories: very good, good, satisfactory and unsatisfactory (Table 4).

### 4. Results and discussion

For a better understanding of the research and subsequent discussion, the results are ranked in tables and figures according to the AIU values. After assessing the six water balance models for the 16 catchments, described in the previous section, we calculated R of the observed and simulated streamflows (Qobs-Qsim) for all evaluated approaches (Table 5). GR2M shows the best fit in most catchments (13 out of 16), both semi-arid and sub-humid, with an average correlation coefficient of close to 0.90. Only the HOY catchment has a correlation coefficient below 0.80 (0.67). Furthermore, the ABCD and AWBM models did not give the best fit for any of the studied catchments, though their values were similar to those using GR2M, especially for humid and sub-humid catchments. All water balance models show correlation coefficients above 0.80 for humid and humid sub-humid catchments. In the dry sub-humid HOY catchment, some values fell below 0.60, especially with the Témez and Thornthwaite-Mather models and RVA. The Témez model, which is widely used in Spain, achieves the best fit only in the



GAR catchment, taking the same value (0.92) as Guo-5p, and similar to the GR2M coefficient (0.90). The more arid the catchment, the lower the average correlation coefficient obtained.

Fig. 4 shows the scatter plot for each catchment obtained using the best model in terms of correlation coefficient. Dispersion is generally greater for the largest streamflows, while the remaining simulated flows appear to have similar order of magnitude.

The perfect fit line (solid line), which indicates that simulated streamflows are identical to observed streamflows, is usually above the estimated regression line (dashed line). Nevertheless, in RVA, flow estimates are below real estimates, and peak flows are more irregularly distributed than the rest of the study catchments, which is characteristic of these latitudes.

The average NSE for each model considering the 16 basins (Fig. 5, upper left) shows good results for all models except Témez, for which the NSE (0.44) is below 0.50. Despite being a five-parameter model, Guo-5p would be only satisfactory according

to the NSE criterion; thus, its value (0.54) is below 0.65. The rest of the models achieve around 0.65 and could be considered good. In contrast, PBIAS (Fig. 5, upper right) shows better results than the Guo-5p and Témez models, the values of which are 0.65% and 4.35%, respectively. Nevertheless, the rest of the models always give below 15% and could be considered good. The Thornthwaite-Mather model is the only "good" model according to RSR criterion (at the bottom left of Fig. 5) and all the others are satisfactory, with the exception of the Témez model (unsatisfactory because its average value is below 0.70).

According to the Bressiani criteria (Fig. 5, bottom right), only the Témez model is unsatisfactory, whereas the rest are, on average, between good and satisfactory.

With regard to the average of the models' results in each basin (Fig. 6), the influence of the aridity index is highlighted. In all basins, from the HOY catchment to the driest catchment, the NSE values (Fig. 6, upper left) are considered unsatisfactory on average, taking into account all the models studied. This same occurs in relation to RSR (Fig. 6 bottom left), with values below

20 0.70 in the HOY catchment and drier ones. Due to the better results obtained for the PBIAS (Fig. 6, upper right) in all catchments except for the TAM catchment, the Bressiani criteria show unsatisfactory results in the four basins that are drier than the JUB catchment and in the HOY catchment (Fig. 6, bottom right).

Table 6 summarizes the classification sum according to the Bressiani criteria for all studied catchments and models. GR2M gives the best fit for nearly 40% of the catchments with values over 7, meaning they all have a very good fit. ABCD model

has very similar results in five of the 16 catchments studied. The Témez and Thornthwaite-Mather models are the best in four and three catchments, respectively. However, the value obtained with the Temez model in the BOL catchment is just 4. AWBM model is the best only in one catchment but shows high values (above 7) in some humid and sub-humid regions. The models that show the best results, on average, in all catchments are ABCD, GR2M and Thornthwaite-Mather, with values around 4.5. AWBM and Témez show the worst results, with 3.65 and 3.00, respectively, which is almost unsatisfactory. Moreover, the

Témez model shows the highest coefficient of variation (129.22%), far from the rest of the models, the average value of which is around 80%. As with previous comparison methods, the best results are obtained in more humid basins, and the drier region shows more unsatisfactory results.

If an ensemble of models is performed (Table 6), the results again depend on the aridity of the region. Humid and sub-humid ensemble models catchments could be classified as very good and good except for the PUE and COT catchments. However,



the results of the dry sub-humid are more irregular, especially in the less arid regions, with classifications from good to unsatisfactory, as in the HOY catchment. The more arid regions (from BOL to RVA) did not show satisfactory results in an ensemble average model.

The main aim of hydrological balance models is to assess inflows in a water resources system, and it is essential for appropriate

analysis of its management. Therefore, in addition to the assessment performed with the Bressiani classification (Table 6), we should analyse the differences between total observed and simulated runoff volumes to validate or discard a model.

Table 7 shows the REV results of these comparisons for the catchments studied and the models considered. According to this criterion, Guo-5p shows the best results, on average, giving the best fitted model in six of the catchments studied. Furthermore, the Guo-5p model is the only water balance model that gives REV values below 15% in semi-arid and dry sub-humid regions,

despite the unsatisfactory results obtained in accordance with the Bressiani criteria (Table 6). The other models give an average REV above 50% in these catchments. With regard to humid and sub-humid catchments, the lowest REV values accord with the Bressiani classification in most cases, which allows for selecting the best model in each catchment (Table 8).

Table 8 shows the proposed model for each catchment, taking into account the proposed set of criteria: R, the Bressiani classification and REV. R does not differ greatly between the various regions, altitudes or areas, although there is a slight

descending trend (from 0.91 to 0.80) when we move to less humid catchments. HOY is the only dry sub-humid catchment where the correlation coefficient was less than 0.60, besides obtaining a sum of only 5 in Bressiani criteria, which is far from an average of 7.5 of its aridity group. This may be because, despite being a small basin (66.15 km$^2$), it is the highest catchment of those studied. Nonetheless, the Thornthwaite-Mather model is classified as good in this catchment, and REV is below 2%. GR2M, The Témez and Guo-5p models are considered the models that best fit in humid regions, though in these regions,

almost all models show good results, with higher percentages in the GR2M model. Thornthwaite-Mather gives the best results in humid sub-humid regions, and Témez gives the worst fit, giving a classification of 'unsatisfactory' in all catchments. However, despite the 'very good' classification, REV percentages in the less humid catchments (ZUM and JUB) are around 20%. The driest and semi-arid catchments did not have 'satisfactory' classifications with any of the studied models, but REV is lower than 10% when the Guo-5p model is used. In contrast, the Témez model 'satisfactory' classification in the BOL

catchment has an overestimate of nearly 30%. No trend was found with catchment area or altitude and the models used.

Fig. 7 shows the comparison between the observed and simulated streamflows in both the calibration (1980–1995) and validation (1995–2010) periods with the selected models in the catchments studied. A qualitative inspection of the graphs shows that the simulated hydrographs are in close agreement with the observed ones. The humid and sub-humid models show good performance in predicted runoff volumes. The validation period is even better fitted than the calibration period in most

30 catchments. Only the highest peaks in runoff discharges in humid and sub-humid regions are underestimated. Sub-humid regions follow a similar pattern, but peaks are overestimated in most catchments (SEG, ZUM, BOL and CUE). The 'unsatisfactory' classification (Table 8) in the driest basins (CUE and RVA) has different graphical results, as seen in Fig. 7. Whereas Guo-5p has a good fit in the CUE catchment for both high and low streamflows, despite an overestimation of 13.61%

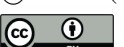



in total runoff volumes, in RVA, peaks are well estimated; however, the water model underestimates runoff volumes in the rest of the study period.

## 5. Summary and conclusions

Spain has a wide climatic variety due to its complex orography and geographic situation. It has the driest and rainiest regions in continental Europe. Indeed, the 16 basins selected as case studies cover a range of aridity index classifications from humid to semi-arid. Lumped hydrological balance models were proved to perform well in humid and sub-humid regions, regardless of the catchments' altitude or area, showing good results in all cases according to the Bressiani rank classification. The driest regions register 'unsatisfactory' performance for the lumped models used, although the estimated runoff volumes with Guo-5p are very similar to the observed ones with differences below 10%, which is even lower than in some dry sub-humid regions. GR2M is the model that generally gives the best fit in Spain, although other models provide slightly better results when a detailed analysis is performed. The more humid the catchment is, the better any water model fits. In the driest regions, it is the opposite. However, despite the poor results according to the Bressiani classification, the Guo-5p model showed low REV, which indicates that it could be used as a good estimator in water management. The Thornthwaite-Mather model fits the best in dry sub-humid regions. The Témez model, widely used in Spain, only performed well in humid regions, as many of the other water balance models did, but it had the worst results in the dry sub-humid region. It only gave the best fit in the BOL catchment, but its REV was nearly +30%, significantly overestimating water resources in the basin, which could consequently lead to inadequate water management.

The usefulness of R when selecting a model is quite low but can be helpful in the analysis of models' weaknesses regarding highest and lowest runoff volumes and extraordinary situations. The Bressiani classification takes into account different comparison criteria (NSE, RSR and PBIAS) to help in the decision-making process when selecting a model. Nevertheless, the assessment of the margin of error in total runoff volume by using REV is also a key index. NSE and RSR leads to ordering and identifying the models that fit better, but PBIAS does not show conclusive results and can even distort the Bressiani classification. The REV criterion assesses both the overestimation and underestimation of the total, which is a key factor in the analysis of water-management problems. The methodology used can be applied in regions with similar case studies to assess more accurately the resources within a system.

*Competing interests*. The authors declare that they have no conflict of interest.

*Acknowledgments*. The authors gratefully acknowledge support from the UCAM University of San Antonio of Murcia through the project "PMAFI/06/14". This research was also partially supported by the GESINHIMPADAPT project (CGL2013-48424-C2-2-R) with Spanish MINECO funds.





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

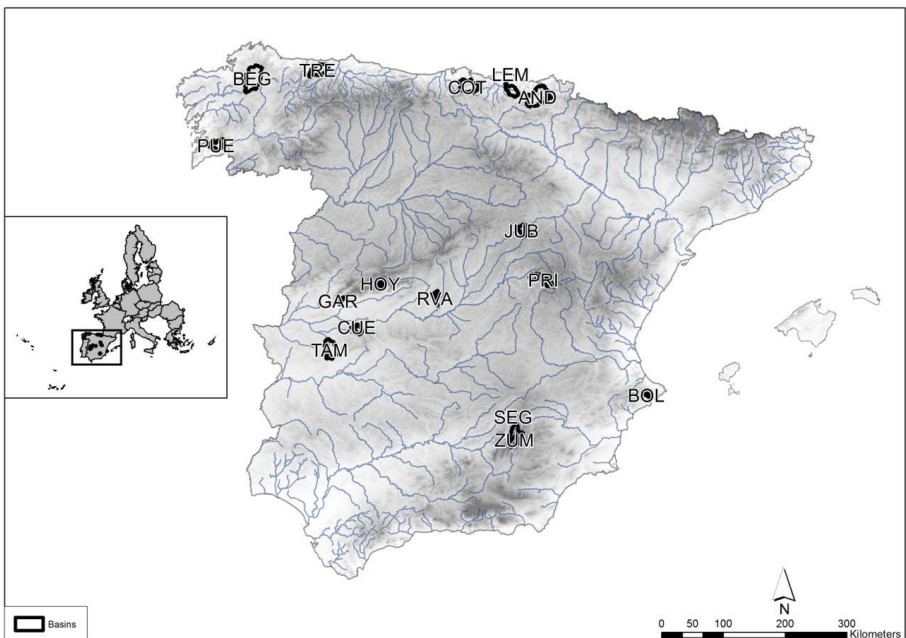

**Fig. 1. Position and location of catchments.**





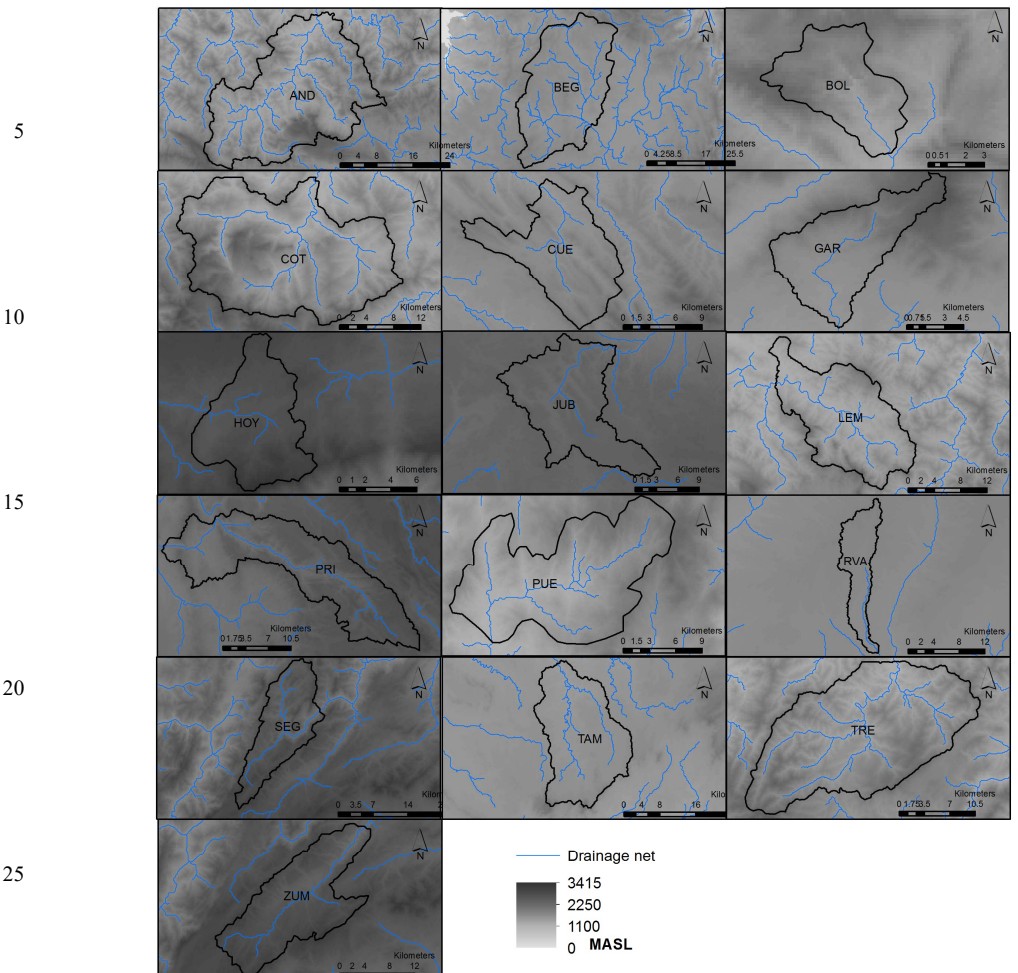

**Fig. 2. Morphology, drainage net and elevation of test catchments.**

30





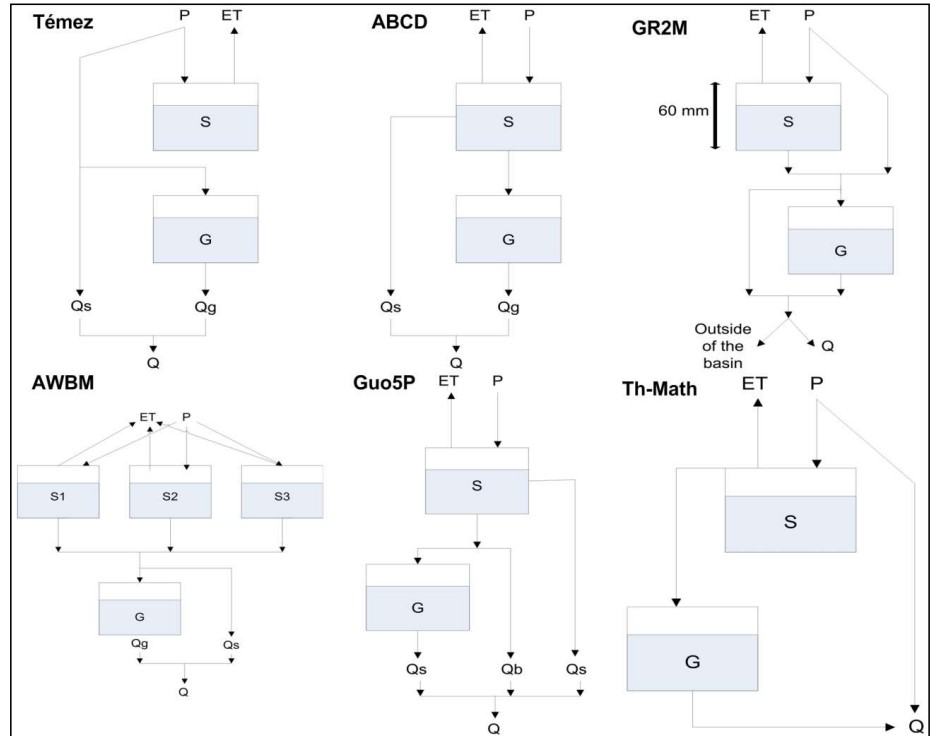

Fig. 3. Conceptual representation of the six water balance models studied.



**Fig. 4. Scatter plots (observed and simulated streamflows in hm3/month) for best model fit according to Pearson's correlation coefficient (R). The dashed line is the estimated regression line, and the solid line is the perfect fit.**



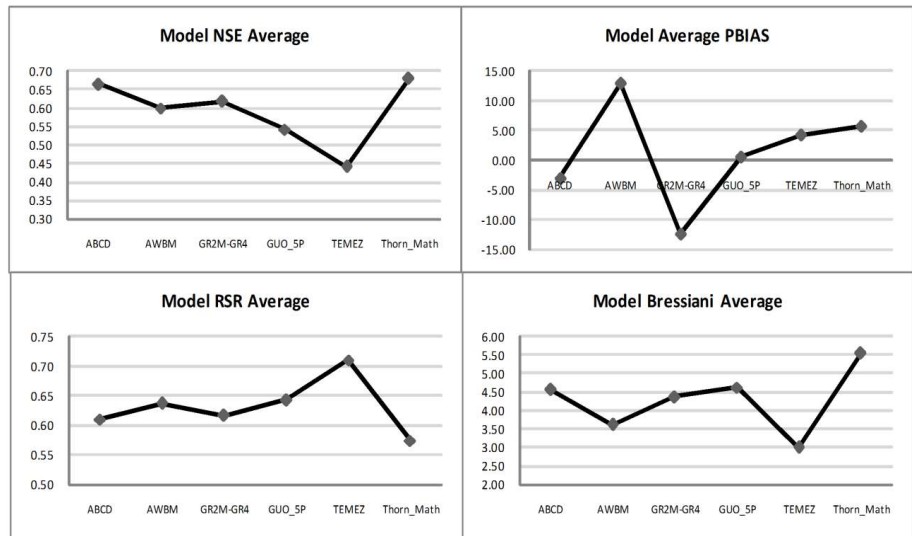

Fig. 5. Model average comparison

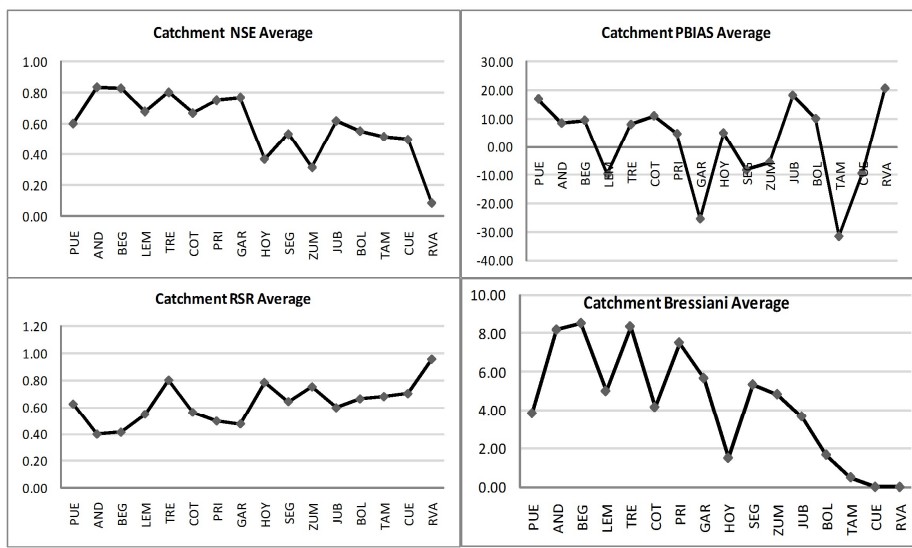

Fig. 6. Comparison of catchment averages



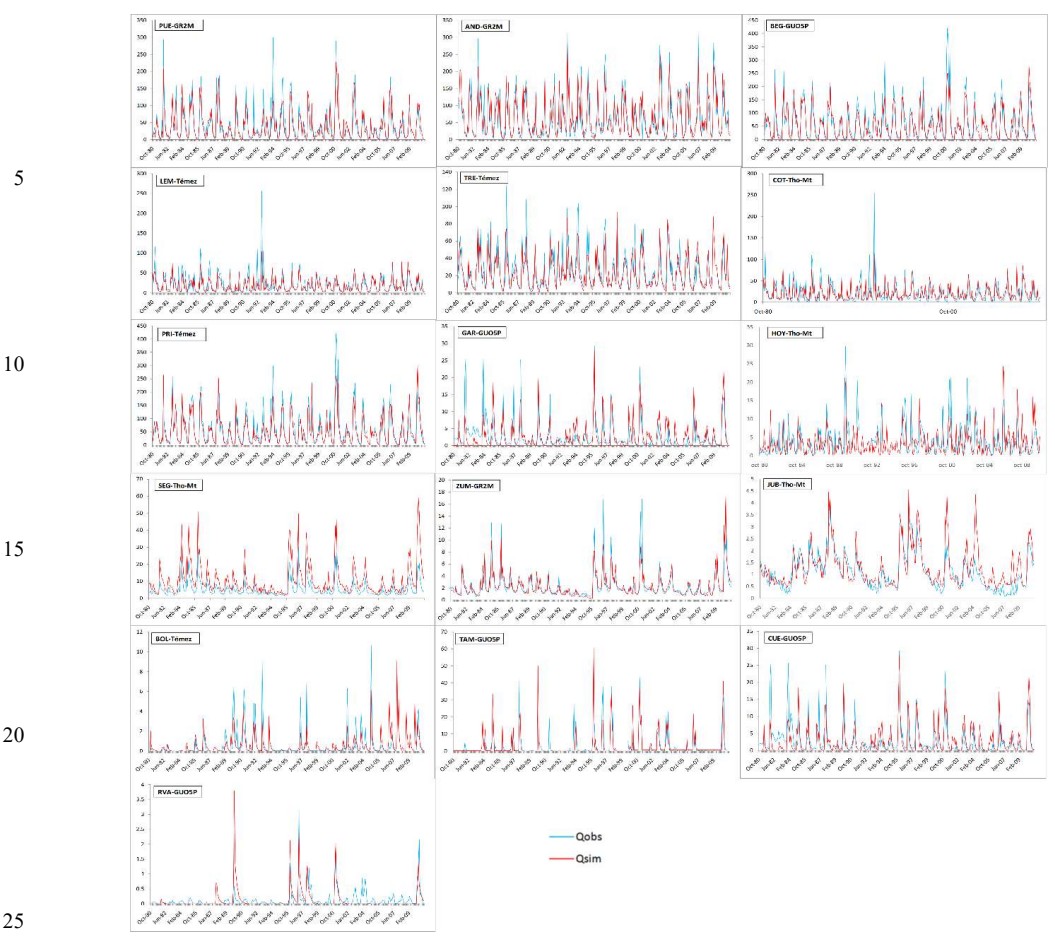

**Fig. 7. Observed and simulated runoff discharge graphs in the study period (1980–2010).**

30

35



**Table 1.** Summary of catchments characteristics (1977–2010) (X and Y coordinates refer to the centroid of the basin.

| Code | Area (km²) | X ETRS89 UTM 30N | Y ETRS 89 UTM 30N | MASL (m) | Köppen Class. | UNEP Aridity Index | Average temperature (ºC) | Average yearly precipitation (mm) | Average yearly ETP (mm) |
|---|---|---|---|---|---|---|---|---|---|
| AND | 778.49 | 573531.02 | 4769234.41 | 486.01 | Cfb | 2.15 | 11.62 | 1563.47 | 727.86 |
| BEG | 836.89 | 111096.28 | 4799111.26 | 504.01 | Csb | 2.07 | 11.44 | 1332.62 | 632.50 |
| BOL | 29.23 | 749970.13 | 4286756.78 | 600.31 | Csa | 0.54 | 16.56 | 579.71 | 1080.30 |
| COT | 488.22 | 460081.35 | 4786345.03 | 559.51 | Cfb | 1.65 | 11.48 | 1311.12 | 793.16 |
| CUE | 139.86 | 280997.89 | 4392588.21 | 610.63 | Csa | 0.52 | 15.33 | 570.56 | 1098.46 |
| GAR | 69.92 | 251972.85 | 4439204.60 | 689.98 | Csa | 1.02 | 14.73 | 1060.18 | 1043.32 |
| HOY | 66.15 | 318316.11 | 4466995.19 | 1632.12 | Csb | 1.01 | 8.58 | 777.22 | 770.35 |
| JUB | 207.66 | 549719.20 | 4553001.74 | 1150.05 | Csb | 0.65 | 10.84 | 509.76 | 783.22 |
| LEM | 252.58 | 530214.27 | 4779163.90 | 342.18 | Cfb | 1.96 | 12.35 | 1393.18 | 709.20 |
| PRI | 328.16 | 578643.32 | 4473678.12 | 1255.05 | Csb | 1.19 | 10.96 | 763.04 | 642.86 |
| PUE | 263.85 | 52975.07 | 4692077.36 | 400.05 | Csb | 2.20 | 14.07 | 1662.05 | 756.42 |
| RVA | 85.68 | 407367.20 | 4444846.50 | 607.70 | Bsk | 0.40 | 14.69 | 396.53 | 998.49 |
| SEG | 232.89 | 533459.57 | 4225568.22 | 1416.46 | Csb | 0.88 | 11.53 | 807.66 | 915.83 |
| TAM | 458.12 | 237068.72 | 4360580.97 | 447.46 | Csa | 0.54 | 15.93 | 596.36 | 1112.84 |
| TRE | 413.54 | 217492.08 | 4812225.19 | 526.64 | Cfb | 1.84 | 12.31 | 1220.91 | 663.77 |
| ZUM | 266.03 | 536758.71 | 4213732.23 | 1549.95 | Csb | 0.79 | 11.35 | 750.28 | 951.88 |

**Table 2.** Classification of the regions according to UNEP aridity index (AIU).

| Classification | UNEP Aridity Index (AI$_U$) |
|---|---|
| Hyper-arid | AI$_U$ <0.05 |
| Arid | 0.05< AI$_U$ <0.20 |
| Semi-arid | 0.20< AI$_U$ <0.50 |
| Dry sub-humid | 0.50< AI$_U$ <0.65 |
| Humid sub-humid | 1.00< AI$_U$ <0.65 |
| Humid | AI$_U$ >1.00 |





**Table 3.** Goodness-of-fit formula tests. Qobs, Vobs and Qsim, Qsim are the observed and simulated streamflow and runoff volumes, respectively, and STDEV is the standard deviation.

| Goodness-of-fit test | Eq. | Value Range | Optimal Value |
|---|---|---|---|
| $NSE = 1 - \dfrac{\sum_{i=1}^{n}\left(Q_{obs,i} - Q_{sim,i}\right)^2}{\sum_{i=1}^{n}\left(Q_{obs,i} - \overline{Q_{obs}}\right)^2}$ | (2) | $-\infty$ to 1 | $> 0.50$ |
| $RMSE = \sqrt{\dfrac{\sum_{i=1}^{n}\left(Q_{sim,i} - Q_{obs,i}\right)^2}{n}}$ | (3) | 0 to $\infty$ | $<$ STDEV/2 |
| $R = \dfrac{S_{obs,sim}}{\sqrt{S_{obs} * S_{sim}}}$ | (4) | | |
| $S_{obs,sim} = \dfrac{1}{n-1} * \sum_{i=1}^{n}\left(Q_{obs,i} - \overline{Q_{obs}}\right) * \left(Q_{sim,i} - \overline{Q_{sim}}\right)$ | (5) | | |
| $S_{obs} = \dfrac{1}{n-1} * \sum_{i=1}^{n}\left(Q_{obs,i} - \overline{Q_{obs}}\right)^2$ | (6) | $-1$ to $+1$ | $-1$ or $+1$ |
| $S_{sim} = \dfrac{1}{n-1} * \sum_{i=1}^{n}\left(Q_{sim,i} - \overline{Q_{sim}}\right)^2$ | (7) | | |
| $PBIAS = \dfrac{\sum_{i=1}^{n}\left(Q_{obs,i} - Q_{sim,i}\right) * 100}{\sum_{i=1}^{n} Q_{obs,i}}$ | (8) | $\pm 100\%$ | $\pm 25\%$ |
| $RSR = \dfrac{RMSE}{STDEV_{obs}} = \dfrac{\sqrt{\sum_{i=1}^{n}\left(Q_{obs,i} - Q_{sim,i}\right)^2}}{\sqrt{\sum_{i=1}^{n}\left(Q_{obs,i} - \overline{Q_{obs}}\right)^2}}$ | (9) | 0 to $\infty$ | $\leq 0.70$ |
| $REV = \dfrac{\sum_{i=1}^{n}\left(V_{obs,i} - V_{sim,i}\right)}{\sum_{j=1}^{n} V_{obs,i}}$ | (10) | $\pm 100\%$ | $\pm 25\%$ |

**Table 4.** Classification criteria for hydrological models (Moriasi et al., 2007; Bressiani et al., 2015).

| | NSE | PBIAS (%) | RSR | Grading | Classification-Sum |
|---|---|---|---|---|---|
| **Very Good (V)** | $0.75 < NSE \leq 1.00$ | PBIAS $< \pm 10$ | $0.00 \leq RSR \leq 0.50$ | 3 | $7 < E \leq 9$ |
| **Good (G)** | $0.65 < NSE \leq 0.75$ | $\pm 10 \leq$ PBIAS $< \pm 15$ | $0.50 < RSR \leq 0.60$ | 2 | $5 < E \leq 7$ |
| **Satisfactory (S)** | $0.50 < NSE \leq 0.65$ | $\pm 15 \leq$ PBIAS $< \pm 25$ | $0.60 < RSR \leq 0.70$ | 1 | $3 < E \leq 4$ |
| **Unsatisfactory (U)** | $NSE \leq 0.50$ | PBIAS $\geq \pm 25$ | $RSR > 0.70$ | Unsatisfactory | Unsatisfactory |





**Table 5.** Correlation coefficient for observed-simulated streamflows.

| Catchment | ABCD | AWBM | GR2M | GUO-5P | Témez | THOR-MATH | Average |
|---|---|---|---|---|---|---|---|
| PUE | 0.79 | 0.84 | **0.91** | 0.83 | 0.84 | 0.77 | 0.86 |
| AND | 0.90 | 0.90 | **0.91** | 0.89 | 0.90 | 0.90 | 0.91 |
| BEG | 0.92 | 0.92 | **0.93** | 0.91 | 0.92 | 0.92 | 0.91 |
| LEM | 0.89 | 0.89 | **0.93** | 0.88 | 0.90 | 0.88 | 0.90 |
| TRE | 0.90 | 0.90 | **0.92** | 0.90 | 0.89 | 0.89 | 0.89 |
| COT | 0.85 | 0.86 | **0.91** | 0.85 | 0.89 | 0.85 | 0.88 |
| PRI | 0.87 | 0.88 | 0.89 | 0.88 | 0.89 | **0.93** | 0.90 |
| GAR | 0.88 | 0.90 | 0.90 | 0.92 | **0.92** | 0.90 | 0.78 |
| HOY | 0.66 | 0.66 | **0.67** | 0.67 | 0.66 | 0.58 | 0.73 |
| SEG | 0.83 | 0.83 | **0.84** | 0.84 | 0.72 | 0.84 | 0.77 |
| ZUM | 0.66 | 0.77 | **0.82** | 0.78 | 0.64 | 0.66 | 0.77 |
| JUB | 0.78 | 0.83 | **0.85** | 0.82 | 0.79 | 0.83 | 0.77 |
| BOL | 0.76 | 0.65 | 0.74 | **0.78** | 0.73 | 0.72 | 0.79 |
| TAM | 0.89 | 0.89 | **0.90** | 0.89 | 0.89 | 0.59 | 0.85 |
| CUE | 0.71 | 0.87 | **0.90** | 0.89 | 0.89 | 0.87 | 0.77 |
| RVA | 0.61 | 0.75 | **0.85** | 0.80 | 0.59 | 0.52 | 0.69 |
| Best fit (Number of times) | **0** | **0** | **13** | **1** | **1** | **1** | |

5    **Table 6.** Bressiani classification values. Best results in bold.

| Catchment | ABCD | AWBM | GR2M | GUO-5P | Témez | THOR-MATH | Average | Ensemble Classification |
|---|---|---|---|---|---|---|---|---|
| PUE | 3 | 0 | **8** | 7 | 0 | 5 | 3.83 | **S** |
| AND | **9** | 7 | **9** | **9** | 7 | 8 | 8.17 | **V** |
| BEG | **9** | 8 | **9** | **9** | 8 | 8 | 8.50 | **V** |
| LEM | 4 | 7 | 0 | 3 | **9** | 7 | 5.00 | **G** |
| TRE | **9** | 7 | **9** | 8 | **9** | 8 | 8.33 | **V** |
| COT | 4 | 0 | 5 | 7 | 0 | **9** | 4.17 | **S** |
| PRI | 7 | 7 | **9** | 7 | 8 | 7 | 7.50 | **G** |
| GAR | 0 | **9** | 7 | **9** | 0 | **9** | 5.67 | **G** |
| HOY | 4 | 0 | 0 | 0 | 0 | **5** | 1.50 | **U** |
| SEG | **9** | 5 | 3 | 4 | 3 | 8 | 5.33 | **G** |
| ZUM | **7** | 5 | **7** | 5 | 0 | 5 | 4.83 | **G** |
| JUB | 5 | 3 | 4 | 3 | 0 | **7** | 3.67 | **S** |
| BOL | 3 | 0 | 0 | 0 | **4** | 3 | 1.67 | **U** |
| TAM | 0 | 0 | 0 | **3** | 0 | 0 | 0.50 | **U** |
| CUE | 0 | 0 | 0 | 0 | 0 | 0 | 0.00 | **U** |



| RVA | 0 | 0 | 0 | 0 | 0 | 0 | 0.00 | U |
|---|---|---|---|---|---|---|---|---|
| **Average** | **4.56** | **3.63** | **4.38** | **4.63** | **3.00** | **4.56** | | |
| **C.V. (%)** | **78.01** | **97.74** | **89.73** | **76.17** | **129.22** | **78.01** | | |
| **Best fit (Number of times)** | **5** | **1** | **6** | **3** | **4** | **4** | | |

**Table 7.** REV (%) values. Best results in bold.

| Catchment | ABCD | AWBM | GR2M | GUO-5P | Témez | THOR-MATH |
|---|---|---|---|---|---|---|
| PUE | -2.95 | -29.46 | 2.67 | -2.57 | -29.98 | **0.60** |
| AND | -10.09 | -15.46 | **1.05** | -3.68 | -15.78 | -13.86 |
| BEG | -5.02 | -10.06 | 21.61 | **0.66** | -8.70 | -9.81 |
| LEM | 16.24 | **-4.49** | -16.22 | -23.47 | 4.62 | 23.55 |
| TRE | 8.20 | -7.55 | 37.68 | 14.06 | **0.74** | -2.86 |
| COT | **-1.20** | -41.41 | 2.19 | -4.34 | -36.84 | 3.43 |
| PRI | **-0.80** | -8.24 | 15.48 | 11.59 | -10.28 | -5.27 |
| GAR | 63.44 | 45.58 | 22.27 | **3.43** | 54.02 | 52.81 |
| HOY | 11.20 | -35.00 | 31.60 | 17.78 | -32.90 | **-1.75** |
| SEG | 20.24 | 5.59 | **-0.75** | -4.07 | 5.43 | 11.21 |
| ZUM | 16.25 | **-3.87** | 16.81 | 6.72 | 26.54 | 7.25 |
| JUB | -18.60 | **-17.92** | 88.28 | 52.69 | -26.40 | -21.58 |
| BOL | 37.20 | 19.25 | -30.62 | **12.39** | 27.70 | 20.38 |
| TAM | 113.54 | 64.59 | -37.41 | **-3.61** | 97.40 | -18.45 |
| CUE | 32.61 | 27.92 | 48.22 | **13.61** | 38.61 | 47.20 |
| RVA | -26.95 | -34.30 | 24.74 | **1.04** | -77.66 | -78.97 |
| **Average (absolute value)** | **24.03** | **23.17** | **24.85** | **10.98** | **30.85** | **19.94** |
| **Best fit (Number of times)** | **2** | **3** | **2** | **6** | **1** | **2** |





**Table 8.** Summary of performance of the selected water models.

| Catchment | Area (km²) | Altitude (MASL) | Water Model | R | Bressiani Sum | Bressiani Classification | REV (%) |
|---|---|---|---|---|---|---|---|
| PUE | 263.85 | 400.05 | GR2M | 0.91 | 8 | Very Good | +2.67 |
| AND | 778.49 | 486.01 | GR2M | 0.91 | 9 | Very Good | +1.05 |
| BEG | 836.89 | 504.01 | Guo5p | 0.91 | 9 | Very Good | +0.66 |
| LEM | 252.58 | 342.18 | Témez | 0.90 | 9 | Very Good | +4.62 |
| TRE | 413.54 | 526.64 | Témez | 0.89 | 9 | Very Good | +0.74 |
| COT | 488.22 | 559.51 | Thor-Math | 0.85 | 9 | Very Good | +3.43 |
| PRI | 328.16 | 1255.05 | Témez | 0.89 | 8 | Very Good | -10.28 |
| GAR | 69.92 | 689.98 | Guo5p | 0.92 | 9 | Very Good | +3.43 |
| HOY | 66.15 | 1632.12 | Thor-Math | 0.58 | 5 | Good | -1.75 |
| SEG | 232.89 | 1416.46 | Thor-Math | 0.84 | 8 | Very Good | +11.21 |
| ZUM | 266.03 | 1549.95 | GR2M | 0.82 | 7 | Very Good | +16.81 |
| JUB | 207.66 | 1150.05 | Thor-Math | 0.85 | 7 | Very Good | -21.58 |
| BOL | 29.23 | 600.31 | Témez | 0.73 | 4 | Satisfactory | +27.70 |
| TAM | 458.12 | 447.46 | Guo5p | 0.89 | 3 | Satisfactory | -3.61 |
| CUE | 139.86 | 610.63 | Guo5p | 0.89 | 0 | Unsatisfactory | +13.61 |
| RVA | 85.68 | 607.70 | Guo5p | 0.80 | 0 | Unsatisfactory | +1.04 |

