# Peer review of "Assessment of lumped hydrological balance models in peninsular Spain"

_Hydrology and Earth System Sciences, 2017_

## Referee Comment (RC1) · Anonymous Referee #1 · 18 Sep 2017

General comments:

The present paper aims to conduct a comparative analysis among six lumped hydrological models applied to streamflow simulation in 16 watersheds in Spain. The watersheds have different climatic regimes with more than 30 years of data. Models are used to generate monthly streamflow, and are compared with respect to six quality metrics. The Bressiani classification scheme is used to assess models performance trying to investigate which model is satisfactory for a given watershed. Even though the paper is well-written, the main problem is not stated well, and the rationale behind using lumped hydrological models (rather than semi-distributed models) is not convincing enough (see specific comments). I believe that this study, rather than a research article, is an exploratory analysis of a number of models, which can be presented in

terms of a technical paper. I also have a number of comments that could improve the quality of this work if authors decide to resubmit the manuscript in future.

Specific comments:

- My main issue with the current study is the fact that authors point out only to a couple of troubles in distributed and semi-distributed modeling, and then decide to use lumped models for streamflow predictions. I argue that there has been a great deal of advances in modeling in the past decades, which has resulted in ease-of-use and high identifiability of semi-distributed models for water management. Lumped models are no longer a justifiable option for water resources modeling. I agree that they can be used for regional analysis, but for watersheds like those studied in this paper, I totally recommend parsimonious semi-distributed models. Using lumped models, even though streamflow can be reproduced satisfactorily at coarse spati-temporal scales (such as monthly, that is considered in this paper), no management decisions could be made regarding water resources. For example, one cannot evaluate the impact of land use change or urbanization on water dynamics. Overall, using lumped models in this study needs to be justified in a better way.

- Regarding the use of lumped models for streamflow predictions, authors say that "The suitability of a model depends on the basin and specific regional characteristics." However, they never justify why lumped models are suitable for the watersheds under study. My impression is that the modeling scheme has been selected just because of simplicity, rather than basin and regional characteristics. Moreover, even though authors list more models in the introduction section, they only use six models. Why?

- It is claimed that this work takes into account the stochastic behaviour of the natural streamflow and the climatic variables. However, nowhere in the paper are any probabilistic analyses employing probability density functions. Just using time series of forcings and streamflow does not mean that the stochastic behavior is taken into account. Moreover, they also say that "We intend to calibrate parsimonious models by

considering the stochastic behaviour of the natural streamflow and climatic variables." However, no details on model calibration (e.g., algorithmic and computational settings) are provided in the manuscript.

- All quality metrics used in this study are aggregate measures that quantify simulations-observations match in an average sense. I believe that, since the ultimate goal in this study is to use model predictions for water management, authors need to utilize more specific quality metrics including those that target low-flows, high-flow, timing, etc. I suggest looking at Gupta and Kling (2009), Yilmaz et al. (2008), and Shafii and Tolson (2015), some examples in a large literature on diagnostic model evaluation.

- Model verification (i.e., in the time frame 1995–2010) has not been demonstrated in the paper. Only Figure 7 graphically shows time series in that time frame, but no quality metrics are calculated and no comparison are made either.

Technical corrections:

- Tables and figures captions need to be longer providing more details. Also, consider merging Figure 1 and 2 in one figure. Fonts in Figure 4 need to be larger. Figure 7 does not demonstrate how well models perform, because it is extremely busy. I recommend using Flow Duration Curve instead of streamflow time series. It makes comparisons conclusions easy to follow.

- On page 7, it is mentioned that "ensemble of models is performed". But there is no information on how it is done. Is it the average of all models, or what?

- Consider rewording the sentence "The range of missing values moves from 2% to 8% in the stations considered" to "Missing values ranges from 2% to 8% in the stations considered"

- Model parameters need to be provided in a table, along with the prior ranges used for model calibration, as well as the optimal values obtained by calibration.

- On page 9, reword "models were proved to perform well" to "models performed well"

- Define "best fit" in Tables 5-7. What conditions need to be satisfied so the model becomes a "best fit" in a watershed?

References: Gupta, H. V., T. Wagener, and Y. Liu (2008), Reconciling theory with observations: elements of a diagnostic approach to model evaluation, Hydrological Processes, 22(18), 3802-3813. Shafii, M., and B. A. Tolson (2015), Optimizing hydrological consistency by incorporating hydrological signatures into model calibration objectives, Water Resources Research, 51(5), 3796-3814. Yilmaz, K. K., H. V. Gupta, and T. Wagener (2008), A process-based diagnostic approach to model evaluation: Application to the NWS distributed hydrologic model, Water Resources Research, 44, W09417.

---

## Referee Comment (RC2) · Anonymous Referee #2 · 10 Nov 2017

This study provides assessment report of a lumped hydrologic models inter-comparison conducted over sixteen mid-sized Spanish River basins. Streamflow simulations of six hydrologic models are compared against observations using six different statistical metrics. The authors reports that the lumped models show better skill in the humid basins as compared to those achieved in the sub-humid and semi-arid basins. They also provide some qualitative guidance on selection of statistical metrics used for model verification. The overall impression I have after reading this work is that it falls quite far way from qualifying as a research article, and as such it appears to me written like a (technical) report.

I have also a hard time to identify novelty or new insights that can be gained from this study – considering that the main finding (models do better in humid catchments) is

well demonstrated in prior works. Another thing that the manuscript seriously lacks is that there is virtually no discussion on why the different models show different behavior – and as such it goes in the direction of showing analysis results to "what" sort of questions and not discussing about "why". How does the selected six hydrologic models differ in terms of other main hydrological fluxes and states (e.g., soil moisture storage and evapotranspiration – besides the aggregated streamflow behavior)?

What is the main motivation behind this model inter-comparison study? I do not have much issue with usage of lumped models, but the authors show clearly motivate their research work - also taking into account why/how did they come up with six models (justification for selected models). As presented now, it appears to me that the authors just came up with six models and then perform the inter-comparison study, without a clear motivation and goal.

I would also suggest authors to tone down the part discussing about lumped and distributed model advantages/limitations. Specifically distributed models have their own advantages, which goes beyond just reproducing the aggregated streamflow dynamics at a catchment outlet. I also do not agree with the author sentence about unavailability of spatial datasets to establish distributed models (line 27, page 2). These days we have now access to lot of spatial datasets (either from remote sensing or ground based) and now it is becoming routinely to apply (fully or semi) distributed models. Please modify your argument.

Page 4, line 1: You mean PET? ET is estimated by hydrologic models. Right?

Section 3.1: Please provide list of calibration parameters and their ranges and optimal values. How did you consider the varying number of calibration parameters among models in your overall ranking process?

Section 3.2: What is the objective function used in the model calibration? How sensitive the model (ranking) results are to the selection of particular objective function?

Section 4: There are only results presented in this section and virtually no discussion, so please modify the section title. How do your results fits to the previous inter-comparison study? Discuss your results in context of previous study – there is not a single citation in this section linking back your results to previous findings.

Page 6, line 25: To which period (calibration or verification) these results belong? Check also elsewhere. Could you present the calibration and validation period results separately? Is the model ranking similar between two periods?

Page 8, line 1: What do you mean by "irregular"?

I find caption of nearly all figures not very informative. For example in Figure 4 – is it for the calibration or verification period? Unit of flow should follow the SI system. Figure 5 and 6 – again to which period, does these results correspond and to which river basins? Figure 7 - please provide indication to calibration and validation periods.

Figure 2: What information is gained by having this figure in the paper - once we have already Fig 1. The quality of Figures 2 and 7 is also very poor. Hard to visualize them and also distinct features could not be identified as presented.

Overall I find there are lots of tables with lots of information inside - that are not thoroughly discussed in the text. Try reducing them to only couple of informative tables and discuss them in detail. Tables like 2 and 3 could be easily dropped out do not add much to the overall message of this paper.

---

## Author Comment (AC1) · 2 Dec 2017

Journal: HESS
Title: Assessment of lumped hydrological balance models in peninsular Spain
Author(s): Julio Pérez-Sánchez et al.
MS No.: hess-2017-424
MS Type: Research article

**Comments from the editors:**

No more referee comments and short comments will be accepted. Now the public discussion shall be completed as follows:

You - as the contact author - are expected to publish final author comments on behalf of all co-authors no later than 08 Dec 2017 (final response phase). You are kindly requested to answer all referee comments and relevant short comments at: https://www.hydrol-earth-syst-sci-discuss.net/hess-2017-424/#discussion

To keep manuscript turnover times low we encourage you to submit your responses as soon as possible. Please note that your revised manuscript should not be prepared at this stage. Based on the responses, the Editor will be asked to take a decision about the further handling of your manuscript.

**Response to Comments from the Editors:**

We appreciate the editor giving us the opportunity to improve the paper during the review process. Following the editor's suggestions, we have replied to each of the reviewers' comments. Based on these answers, if the editor considers it appropriate, we will prepare the revised version of the manuscript.

1) **The present paper aims to conduct a comparative analysis among six lumped hydrological models applied to streamflow simulation in 16 watersheds in Spain. The watersheds have different climatic regimes with more than 30 years of data. Models are used to generate monthly streamflow, and are compared with respect to six quality metrics. The Bressiani classification scheme is used to assess models performance trying to investigate which model is satisfactory for a given watershed. Even though the paper is well-written, the main problem is not stated well, and the rationale behind using lumped hydrological models (rather than semi-distributed models) is not convincing enough (see specific comments). I believe that this study, rather than a research article, is an exploratory analysis of a number of models, which can be presented in terms of a technical paper. I also have a number of comments that could improve the quality of this work if authors decide to resubmit the manuscript in future.**

We thank the reviewer for the recognition of the interest of this research and for the comments formulated, which have helped us to improve the quality of the paper and to focus on the research issues related to the benefits of the use of lumped models in similar areas as the studied ones for water resource management purposes.

2) **My main issue with the current study is the fact that authors point out only to a couple of troubles in distributed and semi-distributed modeling, and then decide to use lumped models for streamflow predictions. I argue that there has been a great deal of advances in modeling in the past decades, which has resulted in ease-of-use and high identifiability of semi-distributed models for water management. Lumped models are no longer a justifiable option for water resources modeling. I agree that they can be used for regional analysis, but for watersheds like those studied in this paper, I totally recommend parsimonious semi-distributed models. Using lumped models, even though streamflow can be reproduced satisfactorily at coarse spatio-temporal scales (such as monthly, that is considered in this paper), no management decisions could be made regarding water resources. For example, one cannot evaluate the impact of land use change or urbanization on water dynamics. Overall, using lumped models in this study needs to be justified in a better way.**

Following this suggestion, we will do our best to improve the justification of the use of lumped models in the new version of the paper. We will modify the paragraph related to the advantages of lumped models as follows:

*With the new development of computer-aided tools and more detailed information, there has been an increasing trend to use distributed or semi-distributed models (Eder et al., 2005; Arnold et al., 1998). They provide more detailed distributed results on a catchment scale approximating heterogeneities of the system.* Following these assumptions, many complex models have been developed that are assumed to be capable of simulating environmental change. However, these spatially explicit and physically based model approaches are often criticized because the necessary a priori estimation of model parameters is difficult (Beven, 2001; Ewen and Parkin, 1996), and u*ncertainty at high resolution may diminish potential gains in prediction accuracy (Carpenter, 2006). Thus, an efficient calibration of the models is difficult due to the spatially distributed nature of those models, showing, sometimes, a decrease in efficiency (Viney et al.,*

*2005), and lumped models can provide a more appropriate alternative (Croke et al., 2004). Moreover, lumped models do not need as much data as distributed models, and the complexity and requirements to process them are lower. Calibration of the lumped parameter models is much less time consuming and produces higher overall model performance in comparison to the more complex distributed models (Vansteenkiste et al., 2014). Despite the simplicity of lumped models, they perform well in many studies (Yang and Michel, 2000; Cameron et al., 1999; Uhlenbrook et al., 1999; Yang et al., 1995). Other studies which have compared lumped and distributed models confirmed that both of them lead to similar accuracy (Lobligeois et al., 2013; Smith et al., 2012; Apip et al., 2012; Breuer et al., 2009; Zhang et al., 2004; Koren et al., 2004; Ajami et al., 2004; Reed et al., 2004; Boyle et al., 2001; Refsgaard and Knudsen, 1996; Shah et al., 1996) and lumped models can be calibrated more efficiently (Bornmann et al., 2009). Vansteenkiste et al. (2014) found out in a Belgium catchment that the lumped models perform better than the distributed ones both in seasonal events and in terms of overall water balance with very low discrepancies in the subflow volumes in comparison to the distributed models. Furthermore, lumped models provide a valuable integrated view of the basin outlet response, as concluded by the Distributed Model Intercomparison Project for the Oklahoma region by Reed et al. (2004) and Smith et al. (2012). Martínez-Santos and Andreu (2010) used lumped and distributed approaches to model natural recharge in semiarid aquifers in Spain, and, though both approaches performed similarly, the lumped models exhibited a better agreement with field records.*

*Therefore, not only spatial discretization determines the quality of the simulation. The choice of model is dictated by the modelling purpose. When flow at the catchment outlet is the main required goal in water resources management, lumped models may be the best choice. But when spatially explicit predictions or land use change predictions are required, a (semi-)distributed model would be more appropriate (Bornmann et al., 2009), though land use changes can be also carried out in lumped model by transfer functions (Breuer et al., 2009). Recently, lumped models have been used, among other purposes, in order to obtain detailed assessments of surface flow, water balance components and the impact of climate change (Haque et al., 2015; Teng et al., 2011; Jiang et al., 2007); to estimate catchment discharge (Kumar et al., 2015; vanEsse et al., 2013; Seiller et al., 2012; Velázquez et al., 2010), to explore transferability under contrasting climate conditions (Broderick et al., 2016, Bourguin et al., 2015) etc.*

NEW REFERENCES:

Ajami, N.K., Gupta, H., Wagener, T., Sorooshian, S. Calibration of a semidistributed hydrologic model for streamflow estimation along a river system. J. Hydrol..298, 112–135, 2004.

Apip, Sayama, T., Tachikawa, Y., Takara, K. Spatial lumping of a distributed rainfall sediment-runoff model and its effective lumping scale. Hydrol.Process.,26, 855–871, 2012.

Beven, K.J. Rainfall-Runoff Modelling, The Primer, Chichester, Wiley, 2001.

Bormann H., Breuer L., Giertz S., Huisman J.A., Viney N.R. Spatially explicit versus lumped models in catchment hydrology – experiences from two case studies. In: Baveye P.C., Laba M., Mysiak J. (eds) Uncertainties in Environmental Modelling and Consequences for Policy Making. NATO Science for Peace and Security Series C: Environmental Security. Springer, Dordrecht, 2009.

Bourgin, F., Andréassian, V., Perrin, C., Oudin, L. Transferring global uncertainty estimates from gauged to ungauged catchments, Hydrol. Earth Syst. Sci., 19, 2535-2546, https://doi.org/10.5194/hess-19-2535-2015, 2015.

Breuer, L., Huisman, J.A., Willems, P., Bormann, H., Bronstert, A., Croke, B.F.W., Frede, H-G., Gräff, T., Hubrechts, L., Jakeman, A.J., Kite, G., Leavesley, G., Lanini, J., Lettenmaier, D.P., Lindström, G., Seibert, J.,Sivapalan, M., and Viney, N.R. Assessing the impact of land use change on hydrology by ensemble modeling (LUCHEM) I: Model intercomparison of current land use.Adv. Water Resour., doi:10.1016/j.advwatres.2008.10.00, 2009.

Broderick, C., Matthews,T., Wilby,R. L., Bastola,S., Murphy, C. Transferability of hydrological models and ensemble averaging methods between contrasting climatic periods, Water Resour. Res., 52, 8343–8373, doi:10.1002/2016WR018850, 2016.

Coff,B.E., Ditty,N.J., Gee,M.C., Szemis,J.M., Maier,H.R., Dandy,G.C., Gibbs,M.S.Relating catchment attributes to parameters of a salt and water balance model.18th World IMACS / MODSIM Congress, Cairns, Australia 13-17 July 2009 http://mssanz.org.au/modsim09, 2009.

Croke, B.F.W., Merritt, W.S., Jakeman, A.J. A dynamic model for predicting hydrologic response to land cover changes in gauged and ungauged catchments, J. Hydrol., 291, 115–131,2004.

Ewen, J., Parkin, G., Validation of catchment models for prediction land use and climate change impact. 1. Method.J. Hydrol., 175, 583–594,1996.

Haque, M,M., Rahman, A., Hagare, D., Kibria, G., Karim, F. Estimation of catchment yield and associated uncertainties due to climate change in a mountainous catchment in Australia. Hydrol.Process., 29, 19, 4339-4349, 2015.

Jiang, T., Chen, Y. Q., Xu, C. Y., Chen, X. H., Chen X., Singh. V.P. Comparison of hydrological impacts of climate change simulated by six hydrological models in the Dongjiang Basin, South China, J. Hydrol., 336, 3–4, 316–333,2007.

Kumar, A., Singh, R., Jena, P.P., Chatterjee, C., Mishra, A. Identification of the best multi-model combination for simulating river discharge.J. Hydrol., 525, 313-325, 2015.

Littlewood, I. G., Croke,B. F. W., Jakeman,A. J., Sivapalan, M. The role of 'top-down' modelling for Prediction in Ungauged Basins (PUB).Hydrol.Process.,17, 8, 1673 - 1679.2003.

Lobligeois, F., Andréassian, V., Perrin, C., Tabary, P., Loumagne, C. When does higher spatial resolution rainfall information improve streamflow simulation? An evaluation on 3620 flood events.Hydrol. Earth Syst. Sci. Discuss. 10, 12485– 12536, 2013.

Martínez-Santos, P., Andreu, J.M. Lumped and distributed approaches to model natural recharge in semiarid karst aquifers. J. Hydrol., DOI: 10.1016/j.jhydrol.2010.05.018, 2010.

Reed, S., Koren, V., Smith, M.B., Zhang, Z., Moreda, F., Seo, D., Dmipparticipants, A.Overall distributed model intercomparison project results. J. Hydrol., 298, 27–60, 2004.

Refsgaard, J.C., van der Sluijs, J., Brown, J., van der Keur, P.A framework for dealing with uncertainty due to model structure error.Adv. Water Resour., 29,1586–1597,2006.

Smith, M.B., Koren, V., Zhang, Z., Zhang, Y., Reed, S.M., Cui, Z., Moreda, F., Cosgrove, B.A., Mizukami, N., Anderson, E.A. Results of the DMIP 2 Oklahoma experiments. J. Hydrol., 418–419, 17–48, 2012.

vanEsse, W. R., Perrin, C., Booij, M. J., Augustijn, D. C. M., Fenicia, F., Kavetski, D., Lobligeois, F. The influence of conceptual model structure on model performance: a comparative study for 237 French catchments.Hydrol.EarthSyst. Sci., 17, 4227-4239, https://doi.org/10.5194/hess-17-4227-2013, 2013

Velázquez, J. A., Anctil, F., Perrin, C. Performance and reliability of multimodel hydrological ensemble simulations based on seventeen lumped models and a thousand catchments.Hydrol. Earth Syst. Sci., 14, 2303-2317, https://doi.org/10.5194/hess-14-2303-2010, 2010.

Viney, N.R., Croke, B.F.W., Breuer, L., Bormann, H., Bronstert, A., Frede, H., Gräff, T., Hubrechts, L., Huisman, J.A., Jakeman, A.J., Kite, G.W., Lanini, J., Leavesley, G., Lettenmaier, D.P., Lindström, G., Seibert, J., Sivapalan M., and Willems, P. Ensemble modelling of the hydrological impacts of land use change, In Zerger, A. and Argent, R.M. (eds.), MODSIM 2005 International Congress on Modelling and Simulation, Modelling and Simulation Society of Australia and New Zealand, 2967–2973,2005.

Vansteenkiste, T., Tavakoli, M., Van Steenbergen, N., De Smedt, F., OkkeBatelaan, Pereira, F., Willems, P.Intercomparison of five lumped and distributed models for catchment runoff and extreme flow simulation. J. Hydrol., 511, 335-349, 2014.

3) **Regarding the use of lumped models for streamflow predictions, authors say that "The suitability of a model depends on the basin and specific regional characteristics." However, they never justify why lumped models are suitable for the watersheds under study. My impression is that the modeling scheme has been selected just because of simplicity, rather than basin and regional characteristics. Moreover, even though authors list more models in the introduction section, they only use six models. Why?**

The choice of the models is mainly based on known performance in different climatic regions, both in Spain and Europe, and structural diversity, i.e., 3 to 6 free parameters and 2 to 4 storage units, and it will be explained at the end of the introduction section in the new version of the paper. However, an inadequate complexity often results in over-parameterization (Ye et al., 1997; Perrin et al., 2001), so the models with too many parameters (more than 6) are excluded from this study.

Thus, the Témez model has been widely used in Spanish catchments (Escriva-Bou et al., 2017, MIMAM, 2000; Cabezas et al., 1999; Estrela et al., 1999) and by the Spanish government in water management (Estrela, 1992). The ABCD model is widely used and proved to have reasonable predictability and has been compared with numerous monthly water balance models, leading to its recommendation (Fernandez et al., 2000; Martinez and Gupta, 2010; Sankarasubramanian and Vogel, 2002; Sankarasubramanian and Vogel, 2003; Thomas, 1981). Vandewiele and Xu (1992) also found that the ABCD model compares favourably with other more recent monthly water balance models in Belgium. Currently, the ABCD model was found to have the best performance in the Segura River basin in southern Spain (Pellícer-Martínez and Martínez Paz, 2015). The GR2M model is widely used in France in all its different versions (Mouelhi et al., 2006). Wriedt and Fayçal (2009) used this model in 492 catchments in Germany, France, Spain and Portugal and obtained high NSE both in the centre and north of Spain and in Central European basins. The AWBM is one of the most widely used rainfall–run-off models in Australia (Boughton, 2004), but nowadays, it is being used in different world catchments (Zhang et al., 2016; Sharifi et al., 2012, 2004), both humid and dry basins, and proving its performance similar

to distributed models (Makungo et al., 2010). The Thornthwaite and Mather water balance model is valid as a water accounting procedure when only reduced information about hydrologic inputs and aquifer characteristics is available, as applied by Peranginangin et al. (2004). It has been successfully used in different water balance research studies in Spain (Barros et al., 2011; Sharifi and Rodríguez, 2002; Estrela and Sahuquillo, 1997; Donker, 1987). Finally, Guo-5p is an adaptation of Thornthwaite and Mather's model with five parameters, so it was chosen in order to compare with the latter. Its use is particularly recommended in humid and semi-humid regions (Xiong and Guo, 1999; Guo, 1995).

REFERENCES:

Barros R, Isidoro D, Aragüés R. Long-term water balances in La Violada irrigation district (Spain): I. Sequential assessment and minimization of closing errors. Agric Water Manag.,102, 1, 35-45, 2011.

Boughton, W. The Australian Water Balance Model. Environ.Modell.Softw., 19, 10, 943-956, DOI10.1016/j.envsoft.2003.10.007, 2004.

Donker, N. WTRBLN, a computer program to calculate water balance.Comput.Geosci.13, 2, 95–122,1987.

Estrela, T., Sahuquillo, A. Modeling the Response of a Karstic Spring at Arteta Aquifer in Spain.Ground Water, 35, 18–24, doi:10.1111/j.1745-6584.1997.tb00055.x, 1997.

Fernandez, W., Vogel, R., Sankarasubramanian, A. Regional calibration of a watershed model.Hydrol. Sci. J., 45, 5, 689-707, 2000.

Gallart, F., Amaxidis, Y.,Botti, P., Cane, G., Castillo, V., Chapman, P.,Froebrich, J....Investigating hydrological regimes and processes in a set of catchments with temporary waters in Mediterranean Europe. HSJ, 53, 3, 618-628, 2008.

Makungo, R., Odiyo, J.O.,Ndiritu,J.G., Mwaka,N. Rainfall-runoff modeling approach for ungauged catchments: a case study of Nzhelele River subquaternary catchment. Phys. Chem. Earth, 35, 596-607, 2010.

Martinez, G.F., Gupta, H.V. Toward improved identification of hydrological models: A diagnostic evaluation of the "abcd" monthly water balance model for the conterminous United States. Water Resour. Res., 46, 8,2010.

Mouelhi, S., Michel, C., Perrin, C., Andreassian, V..Stepwise development of a twoparameter monthly water balance model. J. Hydrol., 318, 200-214. doi:10.1016/j.jhydrol.2005.06.014, 2006.

Pellicer-Martinez, F., Martínez-Paz, J. M. Contrast and transferability of parameters of lumped water balance models in the Segura River Basin (Spain). Water Environ J, 29, 43–50. doi:10.1111/wej.12091, 2015.

Peranginangin, N., Sakthivadievel, R., Scott, N.R., Kendy, E.,Steenhuis, T.S. Water accounting for conjunctive groundwater/surface water management: case ofthe Singkarak-Ombilin River basin, Indonesia. J. Hydrol., 292, 1–22, 2004.

Perrin, C., Michel, C., Andréassian, V. Does a large number of parameters enhance model performance? Comparative assessment of common catchment model structures on 429 catchments.J. Hydrol.,242, 3, 275-301, 2001.

Sankarasubramanian, A., Vogel, R. M. Annual hydroclimatology of the United States, Water Resour. Res., 38, 6, 1083, doi:10.1029/2001WR000619, 2002.

Sankarasubramanian, A., Vogel, R.M.Hydroclimatology of the continental United States. Geophys. Res. Lett., 30, 7,2003.

Sharifi, M. A. , Rodriguez, E. Design and development of a planning support system for policy formulation in water resources rehabilitation: the case of Alcázar De San Juan District in Aquifer 23, La Mancha, Spain. J. Hydroinform., 4, 3, 157-175, 2002.

Sharifi, F.A.S., Safarpour, S., Ayubzadeh, S.A. Evaluation of AWBM 2002 simulation model in 6 Iranian representative catchments. Pajouhesh-Va-Sazandegi, Nat. Resour.,17, 63, 35-42, 2004.

Sharafi, F., Adamowski, J., Barkhordari, J.Saadat,H.Decision Support Tool for Evaluating Changes in Arid andTropical Watersheds. J.Agr.Eng., 49, 2, 2012.

Thomas, H. Improved methods for national water assessment. Report WR15249270, US Water Resource Council, Washington, DC.1981.

Vandewiele, G., Xu, C.-Y.Methodology and comparative study of monthly water balance models 561 in Belgium, China and Burma.J. Hydrol., 134,1-4, 315-347,1992.

Wriedt,G., Bouraoui, F.Towards a General Water Balance Assessment of Europe. Joint Research Centre – Institute for Environment and Sustainability.Luxembourg: Office for Official Publications of the European Communities, 57 pp. 2009.

Ye, W., Bates, B., Viney, N., Sivapalan, M., Jakeman, A., Performance of conceptual rainfall-runoff, models in low-yielding ephemeral catchments. Water Resour. Res., 33, 1, 153-166, 1997.

Zhang, Q., Liu, J., Singh, V. P., Gu, X., and Chen, X. Evaluation of impacts of climate change and human activities on streamflow in the Poyang Lake basin, China. Hydrol.Process., 30, 2562–2576,doi: 10.1002/hyp.10814, 2016.

4) **It is claimed that this work takes into account the stochastic behaviour of the natural streamflow and the climatic variables. However, nowhere in the paper are any probabilistic analyses employing probability density functions. Just using time series of forcings and streamflow does not mean that the stochastic behavior is taken into account. Moreover, they also say that "We intend to calibrate parsimonious models by considering the stochastic behaviour of the natural streamflow and climatic variables." However, no details on model calibration (e.g., algorithmic and computational settings) are provided in the manuscript.**

Thanks to the reviewer's comment, we have realized that we did not express ourselves properly. We did not mean that we wanted to perform a stochastic modelling of the series. We only intended to highlight the stochastic behaviour of the rainfall process and the necessity of using parsimonious approaches to estimate natural streamflow series, which are intended to be representative series of the stochastic variable depending on the rainfall. These series can be used as inputs of management models to assess the operation of water resource systems. In these cases, we need long time series of inflows

(natural streamflows) to take into account the influence of the stochastic behaviours of the hydrology in the reliabilities of the demand supply. We will change this sentence in the new version of the manuscript:

*We intend to calibrate parsimonious approaches to estimate natural streamflow series, and we intend to generate series that represent the stochastic variability of the rainfall process. These series can be used as inputs of management models to assess the operation of water resource systems. In these cases, we need long time series of inflows (natural streamflows) to take into account the influence of the stochastic behaviours of the hydrology in the reliabilities of the demand supply (Pulido-Velazquez et al., 2007, 2008).*

NEW REFERENCES:

Pulido-Velazquez D, Sahuquillo A, Andreu J, Pulido-Velazquez M. 2007. An efficient conceptual model to simulate surface water body-aquifer interaction in Conjunctive Use Management Models. Water Resources Research 43: W07407, doi: 0.1029/2006WR005064

Pulido-Velázquez, D., D. Ahlfeld, J. Andreu and A. Sahuquillo, 2008. Reducing the computational cost of unconfined groundwater flow in conjunctive-use models at basin scale assuming linear behaviour: The case of Adra- Campo de DalíasJournal of Hydrology 353(1-2): 159– 174. doi:10.1016/j.jhydrol.2008.02.006

As for the model calibration concerns, the next paragraph will be included in the new version of the paper in order to provide details of algorithmic and computational settings, as required:

*The calibration of the parameters of all models is carried out by comparing predicted data with observed data. The optimal value for each parameter is the value that minimizes the differences between both flow series and the objective function that minimizes the sum of the square of deviations. The optimization algorithm used is the Generalised Reduced Gradient (Fylstra et al., 1998), which searches for the extreme values of the functions by the generalized reduced gradient algorithm method (Lasdon et al., 1978).*

NEW REFERENCES:

Fylstra, D.,Lasdon, L., Watson, J., Waren, A. Design and use of the Microsoft Excel Solver.Interfaces,28, 5, 29-55, 1998.

Lasdon, L.S.,Waren, A.D., Jain, A., Ratner, M. Design and testing of a generalized reduced gradient code for nonlinear programming. ACM Trans. Math. Softw., 4, 1, 34-50, 1978.

**5) All quality metrics used in this study are aggregate measures that quantify simulations-observations match in an average sense. I believe that, since the ultimate goal in this study is to use model predictions for water management, authors need to utilize more specific quality metrics including those that target low-flows, high-flow, timing, etc. I suggest looking at Gupta and Kling (2009), Yilmaz et al. (2008), and Shafii and Tolson (2015), some examples in a large literature on diagnostic model evaluation.**

Thank you for your recommendation. As the reviewer suggested, flow duration curves (FDCs) will be incorporated in the new version of the paper for the selected water balance models, and specific hydrological metrics will be evaluated according to Yilmaz et al. (2008) and Shafii and Tolson (2015). The signatures that will assess low-flows, high-flows and mid-flows, in both observed and simulated ones, are shown in equations 1, 2 and 3 and will be included in the final version of the paper:

FDC midsegment slope (MS): $\qquad log\ (Q_{m1}) - log\ (Q_{m2})$ $\qquad\qquad$ (1)

where m1 and m2 are the lowest and the highest flow exceedance probabilities within the midsegment of the FDC (0.2 and 0.7, respectively)

FDC high-segment volume (HV): $\qquad \sum_{h=1}^{H} Q_h$ $\qquad\qquad$ (2)

where h = 1, 2, …, H are flow indices located within the high-flow segment probabilities lower than 0.02; H is the index of the maximum flow.

FDC low-segment volume (LV): $\qquad \sum_{l=1}^{L} [\log(Q_l) - \log(Q_L)]$ $\qquad\qquad$ (3)

where I = 1, 2, …, L are the flow indices located within the flow-segment (0.7-1.0 flow exceedance probabilities); L is the index of the minimum flow.

The score for each signature will be calculated as Eq. 4 (Shafii and Tolson, 2015):

$$D_i = \frac{S_i^{obs} - S_i^{sim}}{S_i^{obs}} x100 \qquad\qquad (4)$$

where $D_i$ is the deviation between the signatures of the observed data (obs) and simulated model result (sim).

All the results are shown in the following points.

NEW REFERENCES

Shafii,M., Tolson, B.A. Optimizinghydrological consistency by incorporating hydrological signatures into model calibration objectives. Water Resour. Res., 5, 5, 3796-3814, 2015.

Yilmaz, K. K., Gupta, H. V., Wagener, T. A process-based diagnostic approach to model evaluation: Application to the NWS distributed hydrologic model. Water Resour. Res., 44, W09417, 2008.

6) **Model verification (i.e., in the time frame 1995–2010) has not been demonstrated in the paper. Only Figure 7 graphically shows time series in that time frame, but no quality metrics are calculated and no comparison are made either.**

Figure 7 will be replaced by the flow duration curves (FDCs) of the selected models for each watershed in order to show a better comparison between the observed and simulated flows in all segments (low, high and medium). Likewise, quality signatures, as described above, will be calculated and compared explaining the results obtained, as can be seen in the next point.

7) **Tables and figures captions need to be longer providing more details. Also, consider merging Figure 1 and 2 in one figure. Fonts in Figure 4 need to be larger. Figure 7 does not demonstrate how well models perform, because it is extremely busy. I recommend using Flow Duration Curve instead of streamflow time series. It makes comparisons conclusions easy to follow.**

Thank you very much for your indications. Following both reviewers' considerations, Figure 2 will be deleted in the final paper. Fonts in Figure 4 will be larger, as can be seen in the new version:

[Figure]

**Fig. 4**. Scatter plots (observed and simulated streamflows in hm³/month) for best model fit according to Pearson's correlation coefficient (R). The dashed line is the estimated regression line, and the solid line is the perfect fit.

Figure 7 will be replaced by the flow duration curves of selected models, as follows:

[Figure]

**Fig. 7.** Flow duration curves for observed and selected models' values in the study period (1980–2010).

A new table with FDC metrics will be added, as follows:

**Table 8.** Quality metrics of FDC (MS: Midsegment slope, HV: High-segment Volume, LV: Low-segment volume, $D_i$: Relative difference percentage).

| Catchment | Water Model | Observed | | | Simulated | | | $D_i$ | | |
|---|---|---|---|---|---|---|---|---|---|---|
| | | MS | HV | LV | MS | HV | LV | MS | HV | LV |
| PUE | GR2M | 0.71 | 1878.65 | 67.54 | 0.88 | 1445.11 | 140.45 | -23.9 | 23.0 | -107.9 |
| AND | GR2M | 0.75 | 2239.05 | 164.43 | 0.71 | 1679.94 | 122.85 | 6.0 | 25.0 | 25.3 |
| BEG | Guo5p | 0.76 | 2283.95 | 61.11 | 0.91 | 1595.10 | 119.92 | -19.5 | 30.2 | -96.2 |
| LEM | Témez | 0.70 | 943.00 | 138.90 | 0.76 | 569.64 | 56.39 | -8.6 | 39.6 | 59.4 |
| TRE | Témez | 0.58 | 701.70 | 47.69 | 0.60 | 599.93 | 95.55 | -3.1 | 14.5 | -100.3 |
| COT | Thor-Math | 0.73 | 1725.80 | 110.60 | 0.44 | 1189.46 | 74.69 | 39.5 | 31.1 | 32.5 |
| PRI | Témez | 0.40 | 321.01 | 29.96 | 0.41 | 261.85 | 78.33 | -0.3 | 18.4 | -161.4 |
| GAR | Guo5p | 0.98 | 107.54 | 181.15 | 1.28 | 107.54 | 62.17 | -30.9 | 0.0 | 65.7 |
| HOY | Thor-Math | 0.70 | 147.23 | 72.38 | 0.55 | 136.91 | 157.54 | 21.4 | 7.0 | -117.7 |
| SEG | Thor-Math | 0.36 | 179.70 | 24.21 | 0.57 | 198.59 | 129.37 | -56.1 | -10.5 | -434.5 |
| ZUM | GR2M | 0.35 | 118.50 | 37.60 | 0.42 | 80.15 | 145.56 | -23.0 | 32.4 | -287.2 |
| JUB | Thor-Math | 0.42 | 33.88 | 27.07 | 0.47 | 83.26 | 81.71 | -13.1 | -145.7 | -201.8 |
| BOL | Témez | 1.10 | 45.43 | 115.76 | 1.49 | 25.59 | 176.95 | -35.1 | 43.7 | -52.9 |
| TAM | Guo5p | 3.34 | 283.49 | 9.26 | 0.35 | 217.60 | 0.06 | 89.6 | 23.2 | 99.4 |
| CUE | Guo5p | 0.79 | 168.00 | 278.79 | 1.54 | 140.80 | 227.97 | -96.04 | 16.19 | 18.23 |
| RVA | Guo5p | 1.09 | 12.55 | 37.87 | 6.05 | 14.56 | 93.40 | -455.4 | -16.1 | -146.6 |

Indeed, the final paragraph of Section 4 (Results and discussion) will be replaced by the following one:

*Since the ultimate goal in this study is to use model predictions for water management, flow duration curves (FDCs) for the observed and simulated streamflows for the whole studied period (1980-2010) have been made (Fig. 7) in order to assess quality metrics related to different segments of streamflow series. As has been analysed before, the more humid the watershed, the best performance obtained (almost) regardless of the model used. All the watersheds had a similar pattern, in general terms, except the TAM, CUE and RVA, which were the driest of those studied. The graphics confirm the previous values of Bressiani classification, only showing good performance in the highest volumes. Concerning the humid and sub-humid regions, almost all models had a good performance in probabilities of exceedance lower than 80%, i.e., in high and medium volumes, but both curves were separated the lower the streamflow was. Table 8 quantifies the variabilities identified by FDCs, using the signatures for high, low and midsegment volumes, described before. Deviations in humid and sub-humid watersheds ranged from 0% to*

*30% in both high volumes and midsegment slopes. However, values in low volumes for humid watersheds showed an average of 41% but reached values higher than 100% in PUE, TRE and PRI watersheds (GR2M and Témez models). The best performance was obtained in the AND watershed with values below 25% for all the signatures. Nonetheless, low volumes in most humid watersheds accounted for less than 5% of the total streamflow in the watersheds in the studied period, so their importance is relative for water resources management. The differences in low volumes increased the less humid the watershed was. While high volumes' differences stood at around 15-30%, signatures in low volumes reached to negative 200-400%, overestimating streamflows in spring and summer seasons. As with humid regions, volumes with probability of exceedance above 80% accounted for less than 8% of the total volume in the studied period. On the contrary, semiarid regions showed good performance in high volumes, but both midsegment slopes and low volumes had differences above 100% in most cases, and high volumes in these watersheds accounted for only 25% of the total streamflows in the studied period.*

8) **On page 7, it is mentioned that "ensemble of models is performed". But there is no information on how it is done. Is it the average of all models, or what?**

Yes, it is the average of the all models. The sentence will be changed as follows:

*An ensemble model has been performed using the average of all models (Table 6), showing that the results again depend on the aridity of the region, and there is not a great difference between the single models assessed and the ensemble model.*

9) **Consider rewording the sentence "The range of missing values moves from 2% to 8% in the stations considered" to "Missing values ranges from 2% to 8% in the stations considered"**

Thank you for the recommendation. The sentence will be modified as suggested in the final version of the paper.

10) **Model parameters need to be provided in a table, along with the prior ranges used for model calibration, as well as the optimal values obtained by calibration.**

A new table will be provided in order to show the prior ranges used for model calibration and the optimal values obtained by calibration, as follows:

**Table 4**. Models' characteristics, parameters' value range and optimal values obtained by calibration.

| Model | Number of storages | Number of optimized parameters | Parameters value range | Optimal value range |
|-------|-------------------|-------------------------------|------------------------|---------------------|
| Témez | 2 | 4 | $50 < H < 250$ | $50 < H < 230$ |
| | | | $0.2 < C < 1$ | $0.2 < C < 1$ |
| | | | $10 < I < 150$ | $13 < I < 150$ |
| | | | $0.001 < \alpha < 0.9$ | $0.001 < \alpha < 0.9$ |
| ABCD | 2 | 4 | $0 < a < 1$ | $a = 1$ |
| | | | $5 < b$ | $5 < b < 554$ |

| | | | | |
|---|---|---|---|---|
| | | | 0<c<1 | 0.15<c<0.83 |
| | | | 0<d<1 | 0.015<d<1 |
| GR2M-GR4 | 2 | 4 | 0.6<X1<1.9 | 0.46<X1<1.87 |
| | | | 0.03<X2<18.2 | 0.05<X2<0.94 |
| | | | 100<a | 52<a<462 |
| | | | 0.2<α<0.5 | 0.2<α<0.47 |
| AWBM | 4 | 6 | 50<C<200 | 42<C<92 |
| | | | 0<B<1 | 0.4<B<0.9 |
| | | | 0<K<1 | 0.1<K<0.7 |
| | | | 0<A1<1.5 | 0.01<A1<0.6 |
| | | | 0<A2<1.5 | 0.01<A2<0.4 |
| | | | 0<A3<1.5 | 0.35<A3<1.0 |
| Guo 5p | 2 | 5 | 0<K0<2 | 0.6<K0<2.0 |
| | | | 0<K1<1 | 0<K1<0.5 |
| | | | 0<K2<1 | 0<K2<0.6 |
| | | | 0<C<1 | 0<C<0.7 |
| | | | 0<S | 250<S<1000 |
| Thornthwaite-Mather | 2 | 3 | 0<α<1 | 0.01<α<1 |
| | | | 0<φ | 0.001<φ<256 |
| | | | 0<λ<1 | 0.001<λ<0.92 |

Indeed, in Section 4 (Results and discussion), a new paragraph will be added, as follows:

*Table 4 shows the main characteristics of the models' structure, besides the range of parameters to be calibrated and the optimal ones found in the models and watersheds studied. Nearly all the optimal parameters of the models vary over the entire possible range. This circumstance is probably caused by the adaptation of the different hydrological processes embedded in the models' structure to the wide diversity of the climatic conditions in the regions analysed. Parameters related to storage capacity of the superficial tanks (H in Témez; a in GR2M; φ in Thornthwaite-Mather; A1, A2, A3 in AWBM etc.) have a strong decreasing value trend the drier the watershed is, due to the average soil moisture throughout the year and specific climatic conditions of each watershed. The finding is further confirmed in other parameters, such as maximum soil moisture included in the Guo-5p model in the S parameter, whose values are around 500-1000 mm in humid and sub-humid regions and drop out until 50-70 mm in the driest watersheds, but no tendency is found when PET is summed, as the ABCD model takes into account in parameter b. However, underground storage is only regionally sensitive when aquifer capacity is important, regardless of the climatic location. Moreover, the ABCD model fails to perform the groundwater processes in semiarid regions ($d = 0$), which is of high importance in*

*these watersheds' hydrological water balance. No correlations were found between the parameters and catchment area.*

**11) On page 9, reword "models were proved to perform well" to "models performed well"**

The sentence will be reworded as suggested in the final version of the paper.

**12) Define "best fit" in Tables 5-7. What conditions need to be satisfied so the model becomes a "best fit" in a watershed?**

Thank you for the recommendation. A short comment will be added in the captions of the Tables 5-7 explaining what "best fit" means for each one. Moreover, following the recommendations of reviewer #2, Tables 5-6 have been modified to add a comparison between calibration and validation. Thus, the tables and their captions will be replaced as follows:

**Table 5.** *Correlation coefficient for observed-simulated streamflows (calibration: 1980-1995/validation: 1995-2010). Best fit (highest value for each watershed) in validation period in bold.*

| Catchment | ABCD | AWBM | GR2M | GUO-5P | Témez | THOR-MATH | Average |
|---|---|---|---|---|---|---|---|
| PUE | 0.79/0.79 | 0.82/0.84 | **0.86/0.91** | 0.81/0.83 | 0.83/0.84 | 0.77/0.77 | 0.97/0.86 |
| AND | 0.94/0.90 | 0.94/0.90 | **0.95/0.91** | 0.93/0.89 | **0.95**/0.90 | 0.94/0.90 | 0.94/0.91 |
| BEG | 0.92/0.92 | 0.92/0.92 | **0.93/0.93** | 0.92/0.91 | 0.92/0.92 | 0.92/0.92 | 0.92/0.92 |
| LEM | 0.81/0.89 | 0.80/0.89 | **0.83/0.93** | 0.81/0.88 | 0.82/0.90 | 0.82/0.88 | 0.82/0.90 |
| TRE | 0.90/0.90 | 0.90/0.90 | **0.92/0.92** | 0.90/0.90 | 0.90/0.89 | 0.90/0.89 | 0.90/0.90 |
| COT | 0.87/0.85 | 0.85/0.86 | **0.91/0.91** | 0.87/0.85 | 0.90/0.89 | 0.87/0.85 | 0.88/0.88 |
| PRI | 0.86/0.87 | 0.85/0.88 | **0.88/0.89** | 0.86/0.88 | 0.88/**0.89** | 0.86/0.87 | 0.86/0.88 |
| GAR | 0.90/0.88 | 0.90/0.90 | **0.93**/0.90 | 0.83/0.92 | 0.90/**0.92** | 0.91/0.90 | 0.90/0.90 |
| HOY | 0.79/0.66 | 0.79/0.66 | **0.83/0.67** | 0.81/0.67 | 0.79/0.66 | 0.68/0.58 | 0.78/0.65 |
| SEG | 0.91/0.83 | 0.91/0.83 | **0.95/0.84** | 0.84/0.84 | 0.93/0.72 | 0.93/**0.84** | 0.91/0.82 |
| ZUM | 0.82/0.77 | 0.89/0.77 | **0.92/0.82** | 0.88/0.60 | 0.83/0.64 | 0.82/0.66 | 0.86/0.71 |
| JUB | 0.81/0.78 | 0.88/0.83 | **0.88/0.85** | 0.86/0.82 | 0.85/0.79 | 0.89/0.83 | 0.86/0.82 |
| BOL | 0.81/0.76 | 0.78/0.65 | **0.84**/0.74 | **0.84/0.78** | 0.77/0.73 | 0.79/0.72 | 0.81/0.73 |
| TAM | 0.75/0.89 | **0.76/0.92** | **0.76**/0.90 | 0.70/0.89 | 0.76/0.89 | 0.43/0.59 | 0.69/0.85 |
| CUE | 0.52/0.71 | 0.70/0.87 | **0.70/0.90** | 0.68/0.89 | 0.67/0.89 | 0.69/0.87 | 0.66/0.86 |
| RVA | 0.68/0.61 | 0.67/0.75 | **0.76/0.85** | 0.70/0.80 | 0.66/0.59 | 0.49/0.52 | 0.66/0.69 |
| Best fit | 0/0 | 1/0 | 16/14 | 1/1 | 1/2 | 0/1 | |

Table 6. *Bressiani classification values (calibration: 1980-1995/validation: 1995-2010). Best fit (highest value for each watershed) in bold.*

| Catchment | ABCD | AWBM | GR2M | GUO-5P | Témez | THOR-MATH | Average | Ensemble Classification |
|---|---|---|---|---|---|---|---|---|
| PUE | 5/3 | 0/0 | **7/8** | 3/7 | 0/0 | 3/5 | 3.0/3.8 | S |
| AND | 8/**9** | 8/7 | **9/9** | **9/9** | 8/7 | **9**/8 | 8.5/8.2 | V |
| BEG | **9/9** | 8/8 | **9/9** | **9/9** | 8/8 | 8/8 | 8.5/8.5 | V |
| LEM | 5/4 | 0/7 | **7**/0 | 5/3 | 3/**9** | 5/7 | 4.2/5.0 | G |
| TRE | **9/9** | 7/7 | **9/9** | **9**/8 | 8/**9** | 7/8 | 8.2/8.3 | V |
| COT | 7/4 | 0/0 | **9**/5 | 6/7 | 0/2 | 6/**9** | 4.7/4.5 | S |
| PRI | 7/7 | 7/7 | **9/9** | 7/7 | 6/8 | 7/7 | 7.2/7.5 | G |
| GAR | 7/4 | 6/**9** | **9**/7 | 0/**9** | **9**/0 | 6/**9** | 6.2/6.3 | G |
| HOY | 0/4 | 0/0 | **7**/0 | 4/0 | 0/0 | 0/**5** | 1.8/1.5 | U |
| SEG | 3/**9** | **9**/5 | **9**/3 | **9**/4 | 8/3 | **9**/8 | 7.8/5.3 | G |
| ZUM | 3/**7** | **9**/5 | **9/7** | **9**/5 | 3/0 | 4/5 | 6.2/4.8 | G |
| JUB | 0/5 | **9**/3 | **9**/4 | 7/3 | 6/0 | **9/7** | 6.7/3.7 | S |
| BOL | 5/3 | 0/0 | **7**/0 | 5/0 | 3/**4** | 0/3 | 3.3/1.7 | U |
| TAM | 0/0 | 6/0 | 0/**6** | **9**/3 | 4/0 | 0/0 | 3.2/1.5 | U |
| CUE | 0/0 | 0/0 | 0/0 | 0/0 | 0/0 | 0/0 | 0.0/0.0 | U |
| RVA | 0/0 | 0/0 | **5**/0 | 0/0 | 0/0 | 0/0 | 0.0/0.0 | U |
| Average | 4.2/4.8 | 4.3/3.63 | 7.7/4.8 | 5.7/4.6 | 4.1/3.0 | 4.6/4.6 | | |
| Best fit (Number of times) | 3/5 | 3/1 | 14/7 | 6/3 | 1/3 | 3/5 | | |

Table 7. *REV (%) values in period 1980-2010. Best fit (lowest absolute value for each watershed) in bold.*

| Catchment | ABCD | AWBM | GR2M | GUO-5P | Témez | THOR-MATH |
|---|---|---|---|---|---|---|
| PUE | -2.95 | -29.46 | 2.67 | -2.57 | -29.98 | **0.60** |
| AND | -10.09 | -15.46 | **1.05** | -3.68 | -15.78 | -13.86 |

| | | | | | | |
|---|---|---|---|---|---|---|
| BEG | -5.02 | -10.06 | 21.61 | **0.66** | -8.70 | -9.81 |
| LEM | 16.24 | **-4.49** | -16.22 | -23.47 | 4.62 | 23.55 |
| TRE | 8.20 | -7.55 | 37.68 | 14.06 | **0.74** | -2.86 |
| COT | **-1.20** | -41.41 | 2.19 | -4.34 | -36.84 | 3.43 |
| PRI | **-0.80** | -8.24 | 15.48 | 11.59 | -10.28 | -5.27 |
| GAR | 63.44 | 45.58 | 22.27 | **3.43** | 54.02 | 52.81 |
| HOY | 11.20 | -35.00 | 31.60 | 17.78 | -32.90 | **-1.75** |
| SEG | 20.24 | 5.59 | **-0.75** | -4.07 | 5.43 | 11.21 |
| ZUM | 16.25 | **-3.87** | 16.81 | 6.72 | 26.54 | 7.25 |
| JUB | -18.60 | **-17.92** | 88.28 | 52.69 | -26.40 | -21.58 |
| BOL | 37.20 | 19.25 | -30.62 | **12.39** | 27.70 | 20.38 |
| TAM | 113.54 | 64.59 | -37.41 | **-3.61** | 97.40 | -18.45 |
| CUE | 32.61 | 27.92 | 48.22 | **13.61** | 38.61 | 47.20 |
| RVA | -26.95 | -34.30 | 24.74 | **1.04** | -77.66 | -78.97 |
| Average (absolute value) | 24.03 | 23.17 | 24.85 | 10.98 | 30.85 | 19.94 |
| Best fit (Number of times) | 2 | 3 | 2 | 6 | 1 | 2 |

Reviewer #2 Comments to Author:

1) **This study provides assessment report of a lumped hydrologic models intercomparison conducted over sixteen mid-sized Spanish River basins. Streamflow simulations of six hydrologic models are compared against observations using six different statistical metrics. The authors reports that the lumped models show better skill in the humid basins as compared to those achieved in the sub-humid and semi-arid basins. They also provide some qualitative guidance on selection of statistical metrics used for model verification. The overall impression I have after reading this work is that it falls quite far way from qualifying as a research article, and as such it appears to me written like a (technical) report. I have also a hard time to identify novelty or new insights that can be gained from this study – considering that the main finding (models do better in humid catchments) is well demonstrated in prior works.**

We thank the reviewer for the comments formulated, which have helped us to improve the quality of the paper and to review and rewrite some paragraphs and sections to provide more of a research article approach, as the reviewer suggested. The main changes that will be in the final version are the following:

- **The abstract** will be reworked in order to focus on the importance of the issue and the novelties and worldwide applications of our research, as follows:

**Abstract.** *The assessment of inflows in a water resource system is essential for the appropriate analysis of its management. These inflows can be obtained from hydrological balance models. In this paper, we perform a comparative study of six lumped hydrological balance models in several basins with different climatic conditions within Spain. We have selected basins where long time series of climatic and hydrological data are available (more than 30 years). The study period comprises 34 years (1977–2010). The explored models are Témez, ABCD, GR2M, the Australian water balance model (AWBM), GUO-5 parameters (Guo-5p) and Thornthwaite-Mather. The optimal parameters of the models vary over the entire possible range due to the wide diversity of the climatic conditions in the regions analysed. Storage capacity parameters show a strong decreasing value trend the drier the watershed is. Six statistical indices are applied to compare the results of the models: Nash–Sutcliffe model efficiency coefficient (NSE), root-mean-square deviation (RMSE), Pearson's correlation coefficient (R), percent bias (PBIAS), RMSE-observations standard deviation ratio (RSR) and the relative error between observed and simulated run-off volumes (REV). The results show that although lumped models can be employed in humid and sub-humid regions, the more humid the catchments are, the better the results obtained regardless of the catchments' altitude or area. GR2M is the model that gives the best fit in wet regions, although climatic differences between calibration and validation periods produce some decrease in performance criteria. Témez model provide the worst results in dry sub-humid and semiarid regions. Guo-5p estimates run-off volumes with errors below 10% despite the unsatisfactory results provided according to the Bressiani classification. These results imply that it is not adequate to rely on a single lumped model, but also, the choice depends on the purpose of studies. The Bressiani classification takes into account different comparison criteria to help in the decision-making process when selecting a model. Nevertheless, the assessment of the margin of error in total run-off volume using REV is also a key index. The usefulness of Pearson's correlation when selecting a model is quite low but can be helpful in the analysis of models' weaknesses. Flow duration curves and metrics confirm the previous results showing a good*

*performance in probabilities of exceedance lower than 80% in wet watersheds. Deviations in low streamflows account for less than 5% of the total streamflow in the watersheds, so inaccuracy is admissible for water management purposes.*

- **The introduction** will be modified in terms of justifying the use of lumped models, as reviewer #1 recommended. Likewise, at the end of the introduction section, the choice of the six models selected in the study will be also explained since they are based on known performance in different climatic regions, both in Spain and Europe. New paragraphs will be shown in the following points.

- **Methodology**: As reviewer #1 suggested, flow duration curves (FDCs) will be incorporated in the new version of the paper for the selected water balance models, and specific hydrological metrics will be evaluated according to Yilmaz et al. (2008) and Shafii and Tolson (2015). The signatures that will assess low-flows, high-flows and mid-flows, in both observed and simulated ones, are shown in equations 1, 2 and 3 and will be included in the final version of the paper:

FDC midsegment slope (MS):     $log\,(Q_{m1}) - log\,(Q_{m2})$                               (1)

where m1 and m2 are the lowest and the highest flow exceedance probabilities within the midsegment of the FDC (0.2 and 0.7, respectively)

FDC high-segment volume (HV):          $\sum_{h=1}^{H} Q_h$                               (2)

where h = 1, 2, …, H are flow indices located within the high-flow segment probabilities lower than 0.02; H is the index of the maximum flow.

FDC low-segment volume (LV):          $\sum_{l=1}^{L} [\log(Q_l) - \log(Q_L)]$                               (3)

where l = 1, 2, …, L are the flow indices located within the flow-segment (0.7-1.0 flow exceedance probabilities); L is the index of the minimum flow.

The score for each signature will be calculated as Eq. 4 (Shafii and Tolson, 2015):

$$D_i = \frac{S_i^{obs} - S_i^{sim}}{S_i^{obs}} x100$$                               (4)

where $D_i$ is the deviation between the signatures of the observed data (obs) and simulated model result (sim).

All the results are shown in the following points.

NEW REFERENCES

Shafii,M., Tolson, B.A. Optimizinghydrological consistency by incorporating hydrological signatures into model calibration objectives. Water Resour. Res., 5, 5, 3796-3814, 2015.

Yilmaz, K. K., Gupta, H. V., Wagener, T. A process-based diagnostic approach to model evaluation: Application to the NWS distributed hydrologic model. Water Resour. Res., 44, W09417, 2008.

- **Figures and Tables**: Figure 2 and Tables 2 and 3 will be deleted (Table 3 will be replaced by the equations of the goodness-of-tests used) according to the reviewer's suggestions. Additionally, we will add a table with the list of parameters of each model and their range and optimal values obtained by calibration, and we will replace Figure 7 with the flow

duration curves (FDCs) of selected models and will include a table with the signatures related to high-flows, low-flows and midsegment flows in the FDCs in order to provide more quality comparative metrics in the study.

- **Results and discussions**: The section will be revised and rewritten in order to discuss the results obtained and explain the performance of the models related to their structure and complexity-simplicity, in addition to climatic conditions of the watersheds. Furthermore, the discussion will be linked to previous findings as recommended. See the new proposed text in point 8.

- **The summary and conclusions** will be rewritten according to the changes carried out and will be focused on new findings and relationships with previous ones, as follows:

*5. Summary and conclusions*

*Spain has a wide climatic variety due to its complex orography and geographic situation. It has the driest and rainiest regions in continental Europe. Indeed, the 16 basins selected as case studies cover a range of aridity index classifications, from humid to semiarid. Lumped hydrological balance models were proved to perform well in humid and sub-humid regions, regardless of the catchments' altitude or area, showing good results in all cases according to the Bressiani rank classification. The driest watersheds produced frequent run-off peaks, and, although they have deep soil, low infiltration was produced, and therefore, the models did not achieve good results.*

*Complexity in the models and overparametrization do not guarantee a better performance, and 3-parameters models such as the Thornwaithe-Mather model show better monthly simulations than the Guo-5p. However, although the driest regions registered "unsatisfactory" performance for the lumped models used and alternative methods (such as machine-learning modelling and semi-distributed models, such as SWAT, 2012) will be assessed in future studies, the estimated run-off volumes with the Guo-5p are very similar to the observed ones with differences below 10%, which is even lower than in some dry sub-humid regions.*

*GR2M is the model that gave the best fit in Spain in the calibration period, but other models provided slightly better results in the validation period according to the Bressiani classification. This is mainly due to the climatic differences between these two periods, especially caused by flash flows that occurred in many watersheds in the last fifteen years. Nevertheless, the performance indexes proved satisfactory in the validation period for most humid and sub-humid watersheds, so this model can be used for future water management in climate change scenarios and on ungauged catchments similar to these basins. The more humid the catchment is, the better any water model fits. In the driest regions, it is the opposite, corroborating previous findings. However, despite the poor results according to the Bressiani classification, the Guo-5p model showed low REV, which indicates that it could be used as a good estimator in yearly run-off. The Thornthwaite-Mather model fit the best in dry sub-humid regions. The Témez model, widely used in Spain, only performed well in humid regions, as many of the other water balance models did, but it had the worst results in the dry sub-humid region. It only gave the best fit in the BOL catchment, but its REV was nearly +30%, significantly overestimating water resources in the basin, which could consequently lead to inadequate water management. These results indicate that it is not adequate to rely on a single lumped model, but also, the choice depends on the purpose of studies. The average ensemble model did not improve the previous single ranking despite the findings by Broderick et al. (2016) in Ireland.*

*The usefulness of R when selecting a model is quite low, but it can be helpful in the analysis of models' weaknesses regarding highest and lowest run-off volumes and extraordinary situations. The Bressiani classification takes into account different comparison criteria (NSE, RSR and PBIAS) to help in the decision-making process when selecting a model. Nevertheless, the assessment of the margin of error in total run-off volume by using REV is also a key index. NSE and RSR lead to ordering and identifying the models that fit better, but PBIAS does not show conclusive results and can even distort the Bressiani classification. The REV criterion assesses both the overestimation and underestimation of the total, which is a key factor in the analysis of water-management problems. FDCs and their metrics confirm the previous results, as the humid and sub-humid regions models have a good performance in probabilities of exceedance lower than 80%, i.e., in high and medium volumes, but both curves are separated the lower the streamflow is. Deviations in these watersheds range from 0% to 30% in both high volumes and the midsegment slope but reach values higher than 100% in low streamflows. Nonetheless, low volumes in most humid watersheds account for less than 5% of the total streamflow in the watersheds, so inaccuracy is admissible for water-management purposes. On the contrary, semiarid regions just show good performance in high volumes, which account for only 25% of the total streamflows in the studied period, disabling the models selected.*

*The methodology used can be applied in regions with similar case studies to assess more accurately the resources within a system.*

NEW REFERENCES

Broderick, C., Matthews, T., Wilby, R.L., Bastola, S., Murphy,C. Transferability of hydrological models and ensemble averaging methods between contrasting climatic periods.Water Resour. Res., 52, 10, 8343-8373, 2016.

Neitsch, S., Arnold, J.,Kiniry, J.E.A., Srinivasan, R., Williams, J. Soil and water assessment tool user'smanual: Version 2012. Available online:http://swat.tamu.edu/documentation/2012-io/

2) **Another thing that the manuscript seriously lacks is that there is virtually no discussion on why the different models show different behavior – and as such it goes in the direction of showing analysis results to "what" sort of questions and not discussing about "why". How does the selected six hydrologic models differ in terms of other main hydrological fluxes and states (e.g., soil moisture storage and evapotranspiration – besides the aggregated streamflow behavior)?**

The Discussion section will be modified in the final version of the paper, taking into account the behaviour of the different selected models according to their structures and comparing their performance related to their parameters and their particular relevance in the metrics used. The new text proposed for the Discussion section is shown in point 8.

3) **What is the main motivation behind this model inter-comparison study? I do not have much issue with usage of lumped models, but the authors show clearly motivate their research work - also taking into account why/how did they come up with six models (justification for selected models). As presented now, it appears to**

**me that the authors just came up with six models and then perform the inter-comparison study, without a clear motivation and goal.**

Our main motivation in this paper is to use model predictions for water management. The first step, which is developed in this paper, addresses the issue of comparing and selecting, using different metrics (flow duration curves will be added in the final version), the lumped water balance model which has the best fit according to our goal: water management. Once we assess and select the best lumped models, we will compare them with other models we are now working on in peninsular Spain: the semi-distributed model SWAT and machine-learning modelling. We will compare them in order to combine these methods to achieve the best performance in different climatic areas. Finally, we will evaluate the impact of climate change in water resources management with the model built and assess the uncertainties of the use of the different data sources as inputs related to water balance models.

In the final version of the paper, we will explain our goals, as mentioned above, in order to focus the main objectives of this paper. Likewise, the choice of the models is mainly based on known performance in different climatic regions, both in Spain and Europe, and structural diversity, i.e., 3 to 6 free parameters and 2 to 4 storage units, and it will be explained at the end of the introduction section in the new version of the paper. However, inadequate complexity often results in overparametrization (Ye et al., 1997; Perrin et al., 2001), so the models with too many parameters (more than 6) are excluded from this study.

**4) I would also suggest authors to tone down the part discussing about lumped and distributed model advantages/limitations. Specifically distributed models have their own advantages, which goes beyond just reproducing the aggregated streamflow dynamics at a catchment outlet. I also do not agree with the author sentence about unavailability of spatial datasets to establish distributed models (line 27, page 2). These days we have now access to lot of spatial datasets (either from remote sensing or ground based) and now it is becoming routinely to apply (fully or semi) distributed models. Please modify your argument.**

Following reviewer #1's suggestions, we will do our best to improve the justification of the use of lumped models in the new version of the paper. We will modify the paragraph related to lumped models as follows:

Thus, the Témez model has been widely used in Spanish catchments (Escriva-Bou et al., 2017, MIMAM, 2000; Cabezas et al., 1999; Estrela et al., 1999) and by the Spanish government in water management (Estrela, 1992). The ABCD model is widely used and proved to have reasonable predictability and has been compared with numerous monthly water balance models, leading to its recommendation (Fernandez et al., 2000; Martinez and Gupta, 2010; Sankarasubramanian and Vogel, 2002; Sankarasubramanian and Vogel, 2003; Thomas, 1981). Vandewiele and Xu (1992) also found that the ABCD model compares favourably with other more recent monthly water balance models in Belgium. Currently, the ABCD model was found to have the best performance in the Segura River basin in southern Spain (Pellícer-Martínez and Martínez Paz, 2015). The GR2M model is widely used in France in all its different versions (Mouelhi et al., 2006). Wriedt and Fayçal (2009) used this model in 492 catchments in Germany, France, Spain and Portugal and obtained high NSE both in the centre and north of Spain and in Central European basins. The AWBM is one of the most widely used rainfall–run-off models in Australia (Boughton, 2004), but nowadays, it is being used in different world catchments (Zhang et al., 2016; Sharifi et al., 2012, 2004), both humid and dry basins, and proving its performance similar to distributed models (Makungo et al., 2010). The Thornthwaite and Mather water balance model is valid as a water accounting procedure when only reduced information about hydrologic inputs and aquifer characteristics is available, as applied by Peranginangin et al. (2004). It has been successfully used in different water balance research studies in Spain (Barros et al., 2011;

Sharifi and Rodríguez, 2002; Estrela and Sahuquillo, 1997; Donker, 1987). Finally, Guo-5p is an adaptation of Thornthwaite and Mather's model with five parameters, so it was chosen in order to compare with the latter. Its use is particularly recommended in humid and semi-humid regions (Xiong and Guo, 1999; Guo, 1995).

REFERENCES:

Barros R, Isidoro D, Aragüés R. Long-term water balances in La Violada irrigation district (Spain): I. Sequential assessment and minimization of closing errors. Agric Water Manag.,102, 1, 35-45, 2011.

Boughton, W. The Australian Water Balance Model. Environ.Modell.Softw., 19, 10, 943-956, DOI10.1016/j.envsoft.2003.10.007, 2004.

Donker, N. WTRBLN, a computer program to calculate water balance.Comput.Geosci.13, 2, 95–122,1987.

Estrela, T., Sahuquillo, A. Modeling the Response of a Karstic Spring at Arteta Aquifer in Spain.Ground Water, 35, 18–24, doi:10.1111/j.1745-6584.1997.tb00055.x, 1997.

Fernandez, W., Vogel, R., Sankarasubramanian, A. Regional calibration of a watershed model.Hydrol. Sci. J., 45, 5, 689-707, 2000.

Gallart, F., Amaxidis, Y.,Botti, P., Cane, G., Castillo, V., Chapman, P.,Froebrich, J....Investigating hydrological regimes and processes in a set of catchments with temporary waters in Mediterranean Europe. HSJ, 53, 3, 618-628, 2008.

Makungo, R., Odiyo, J.O.,Ndiritu,J.G., Mwaka,N. Rainfall-runoff modeling approach for ungauged catchments: a case study of Nzhelele River subquaternary catchment. Phys. Chem. Earth, 35, 596-607, 2010.

Martinez, G.F., Gupta, H.V. Toward improved identification of hydrological models: A diagnostic evaluation of the "abcd" monthly water balance model for the conterminous United States. Water Resour. Res., 46, 8,2010.

Mouelhi, S., Michel, C., Perrin, C., Andreassian, V..Stepwise development of a twoparameter monthly water balance model. J. Hydrol., 318, 200-214. doi:10.1016/j.jhydrol.2005.06.014, 2006.

Pellicer-Martinez, F., Martínez-Paz, J. M. Contrast and transferability of parameters of lumped water balance models in the Segura River Basin (Spain). Water Environ J, 29, 43–50. doi:10.1111/wej.12091, 2015.

Peranginangin, N., Sakthivadievel, R., Scott, N.R., Kendy, E.,Steenhuis, T.S. Water accounting for conjunctive groundwater/surface water management: case ofthe Singkarak-Ombilin River basin, Indonesia. J. Hydrol., 292, 1–22, 2004.

Perrin, C., Michel, C., Andréassian, V. Does a large number of parameters enhance model performance? Comparative assessment of common catchment model structures on 429 catchments.J. Hydrol.,242, 3, 275-301, 2001.

Sankarasubramanian, A., Vogel, R. M. Annual hydroclimatology of the United States, Water Resour. Res., 38, 6, 1083, doi:10.1029/2001WR000619, 2002.

Sankarasubramanian, A., Vogel, R.M.Hydroclimatology of the continental United States. Geophys. Res. Lett., 30, 7,2003.

Sharifi, M. A. , Rodriguez, E. Design and development of a planning support system for policy formulation in water resources rehabilitation: the case of Alcázar De San Juan District in Aquifer 23, La Mancha, Spain. J. Hydroinform., 4, 3, 157-175, 2002.

Sharifi, F.A.S., Safarpour, S., Ayubzadeh, S.A. Evaluation of AWBM 2002 simulation model in 6 Iranian representative catchments. Pajouhesh-Va-Sazandegi, Nat. Resour.,17, 63, 35-42, 2004.

Sharafi, F., Adamowski, J., Barkhordari, J.Saadat,H.Decision Support Tool for Evaluating Changes in Arid andTropical Watersheds. J.Agr.Eng., 49, 2, 2012.

Thomas, H. Improved methods for national water assessment. Report WR15249270, US Water Resource Council, Washington, DC.1981.

Vandewiele, G., Xu, C.-Y.Methodology and comparative study of monthly water balance models 561 in Belgium, China and Burma.J. Hydrol., 134,1-4, 315-347,1992.

Wriedt,G., Bouraoui, F.Towards a General Water Balance Assessment of Europe. Joint Research Centre – Institute for Environment and Sustainability.Luxembourg: Office for Official Publications of the European Communities, 57 pp. 2009.

Ye, W., Bates, B., Viney, N., Sivapalan, M., Jakeman, A., Performance of conceptual rainfall-runoff, models in low-yielding ephemeral catchments. Water Resour. Res., 33, 1, 153-166, 1997.

Zhang, Q., Liu, J., Singh, V. P., Gu, X., and Chen, X. Evaluation of impacts of climate change and human activities on streamflow in the Poyang Lake basin, China. Hydrol.Process., 30, 2562–2576,doi: 10.1002/hyp.10814, 2016.

**5) Page 4, line 1: You mean PET? ET is estimated by hydrologic models. Right?**

Yes, there was a mistake. ET will be replaced by PET in the final version of the paper.

**6) Section 3.1: Please provide list of calibration parameters and their ranges and optimal values. How did you consider the varying number of calibration parameters among models in your overall ranking process?**

The number of parameters of each model was not considered in the ranking process due to the low range that goes from 3 to 6, which does not imply a significant variability in model structure.

A new table will be provided in order to show the parameters in each model, their ranges and the optimal values obtained by calibration, as follows:

**Table 4**. Models' characteristics, parameters' value range and optimal values obtained by calibration.

| Model | Number of storages | Number of optimized parameters | Parameters value range | Optimal value range |
|-------|-------|-------|-------|-------|
| Témez | 2 | 4 | 50<H<250 | 50<H<140 |

| | | | | |
|---|---|---|---|---|
| | | | 0.2<C<1 | 0.2<C<1 |
| | | | 10<I<150 | 13<I<150 |
| | | | 0.001<α<0.9 | 0.2<α<0.9 |
| ABCD | 2 | 4 | 0<a<1 | a=1 |
| | | | 5<b | 157<b<554 |
| | | | 0<c<1 | 0.35<c<0.83 |
| | | | 0<d<1 | 0.015<d<1 |
| GR2M-GR4 | 2 | 4 | 0.6<X1<1.9 | 0.46<X1<1.87 |
| | | | 0.03<X2<18.2 | 0.07<X2<0.94 |
| | | | 100<a | 126<a<462 |
| | | | 0.2<α<0.5 | 0.2<α<0.41 |
| AWBM | 4 | 6 | 50<C<200 | 42<C<92 |
| | | | 0<B<1 | 0.4<B<0.5 |
| | | | 0<K<1 | 0.3<K<0.61 |
| | | | 0.5<A1<1.5 | 0.2<A1<0.3 |
| | | | 0.5<A2<1.5 | 0.35<A2<0.4 |
| | | | 0.5<A3<1.5 | 0.35<A3<0.4 |
| Guo 5p | 2 | 5 | 0<K0<2 | 0.6<K0<1.5 |
| | | | 0<K1<1 | 0<K1<0.5 |
| | | | 0<K2<1 | 0<K2<0.6 |
| | | | 0<C<1 | 0<C<0.7 |
| | | | 0<S | 250<S<1000 |
| Thornthwaite-Mather | 2 | 3 | 0<α<1 | 0.02<α<1 |
| | | | 0<φ | 0.001<φ<256 |
| | | | 0<λ<1 | 0.001<λ<0.91 |

Indeed, in Section 4 (Results and discussion) a new paragraph will be added, as shown in point 8.

**7) Section 3.2: What is the objective function used in the model calibration? How sensitive the model (ranking) results are to the selection of particular objective function?**

In this study, there is no sensitive analysis of the results. As we mentioned above, uncertainties will be explored in in future research.

As the objective function is concerned, the next paragraph will be included in the new version of the paper in order to provide details of algorithmic and computational settings:

*The calibration of the parameters of all models is carried out by comparing predicted data with observed data. The optimal value for each parameter is the value that minimizes the differences between both flow series and the objective function that minimizes the sum of the square of deviations. The optimization algorithm used is the Generalised Reduced Gradient (Fylstra et al., 1998), which searches for the extreme values of the functions by the generalized reduced gradient algorithm method (Lasdon et al., 1978).*

NEW REFERENCES:

Fylstra, D., Lasdon, L., Watson, J., Waren, A. Design and use of the Microsoft Excel Solver.Interfaces, 28, 5, 29-55, 1998.

Lasdon, L.S., Waren, A.D., Jain, A., Ratner, M. Design and testing of a generalized reduced gradient code for nonlinear programming. ACM Trans. Math. Softw., 4, 1, 34-50, 1978.

8) **Section 4: There are only results presented in this section and virtually no discussion, so please modify the section title. How do your results fits to the previous intercomparison study? Discuss your results in context of previous study – there is not a single citation in this section linking back your results to previous findings.**

This section will be deeply revised and modified in order to discuss the results obtained and to establish the relationships between models' structure and their performance. Furthermore, several citations will be added linking previous findings related to issues studied, as follows:

**4. Results and discussion**

*Table 4 shows the main characteristics of the models' structure, besides the range of parameters to be calibrated and the optimal ones found in the models and watersheds studied. Nearly all the optimal parameters of the models vary over the entire possible range. This circumstance is probably caused by the adaptation of the different hydrological processes embedded in the models' structure to the wide diversity of the climatic conditions in the regions analysed. Parameters related to storage capacity of the superficial tanks (H in Témez; a in GR2M; φ in Thornthwaite-Mather; A1, A2, A3 in AWBM etc.) have a strong decreasing value trend the drier the watershed is, due to the average soil moisture throughout the year and specific climatic conditions of each watershed. The finding is further confirmed in other parameters, such as maximum soil moisture included in the Guo-5p model in the S parameter, whose values are around 500-1000 mm in humid and sub-humid regions and drop out until 50-70 mm in the driest watersheds, but no tendency is found when PET is summed, as the ABCD model takes into account in parameter b. However, underground storage is only regionally sensitive when aquifer capacity is important, regardless of the climatic location. Moreover, the ABCD model fails to perform the groundwater processes in semiarid regions (d = 0), which is of high importance in these watersheds' hydrological water balance. No correlations were found between the parameters and catchment area.*

*For a better understanding of the research and subsequent discussion, the results are ranked in tables and figures according to the AIU values. After assessing the six water-balance models for the 16 catchments, described in the previous section, we calculated R of the observed and*

*simulated streamflows (Qobs-Qsim) for all evaluated approaches (Table 5). The GR2M showed the best fit in nearly all catchments both in calibration (16 out of 16) and validation (14 out of 16), both semiarid and sub-humid, with an average correlation coefficient of close to 0.90. Furthermore, the ABCD and AWBM models did not give the best fit in the validation period for any of the studied catchments, though their values were similar to those using the GR2M, especially for humid and sub-humid catchments. Increasing the number of parameters does not guarantee a better performance, as previous studies' findings show (Bai et al., 2015; Perrin et al., 2001; Yew Gan et al. 1997; Michaud and Sorooshian, 1994; WMO, 1975), since the AWBM (6 parameters) did not show better results than the rest, even when it was compared to the lowest parameters model, the Thornthwaite-Mather (3 parameters), which had better results for nearly all watersheds, both humid and semiarid. All water balance models showed correlation coefficients above 0.80 for humid and sub-humid catchments. In the dry sub-humid HOY and semiarid RVA catchments, some values fell below 0.60, especially with the Thornthwaite-Mather and Témez models, despite the latter being widely used in Spain. It achieved the best fit only in the GAR and PRI catchments, taking similar values as the Guo-5p and GR2M coefficient. The more arid the catchment was, the lower the average correlation coefficient obtained was, which is in line with other studies (van Esse et al., 2013; Caron et al., 2012; Bai et al., 2015). Nevertheless, R does not seem to be a good basis to assess the efficiency of the model since the results did not differ much from the others, besides considering, for most watersheds, that nearly all models showed good performance, when in fact they did not when using other goodness-of-fit tests and different criteria, both graphical and metrics.*

*Fig. 4 shows the scatter plot for each catchment obtained using the best model in terms of correlation coefficient. Dispersion was generally greater for the largest streamflows, while the remaining simulated flows appeared to have similar orders of magnitude. The perfect fit line (solid line), which indicates that simulated streamflows were identical to observed streamflows, was usually above the estimated regression line (dashed line), meaning a general underestimation of streamflows with the models selected, though this fact is relatively subjective due to higher density in low flow zones in these graphics. Nevertheless, in RVA, flow estimates were below real estimates, and peak flows were more irregularly distributed than the rest of the study catchments, which is characteristic of these latitudes.*

*The average NSE for each model considering the 16 basins (Fig. 5, upper left) showed good results for all models except Témez, for which the NSE (0.44) was below 0.50. Despite being a five-parameter model, the Guo-5p would only be satisfactory according to the NSE criterion. However, a 3-parameter model such as the Thornthwaite-Mather achieved the best results in average of NSE, taking into account both wet and dry Spain, which confirms the absence of the need of very complex water balance models, as suggested by Clark et al. (2008), Perrin et al. (2001), Jakeman and Hornberger (1993),* Michaud and Sorooshian (1994) or *Beven (1989). The rest of the models achieved around 0.65 and could be considered good. In contrast, the PBIAS (Fig. 5, upper right) showed better results than the Guo-5p and Témez models, the values of which were 0.65% and 4.35%, respectively. Nevertheless, the rest of the models always gave below 15% and could be considered good, so the PBIAS should be considered as a complementary criterion and should be used in conjunction with other goodness-of-fit tests, as studied in this paper. The Thornthwaite-Mather model was the only "good" model according to the RSR criterion (at the bottom left of Fig. 5), and all the others were satisfactory, with the exception of the Témez model (unsatisfactory because its average value was below 0.70). This runs against what Alley (1984) showed for the Thornthwaite-Mather model, which showed good performance in annual flows but not in simulated monthly flows. According to the Bressiani criteria (Fig. 5,*

*bottom right), only the Témez model was unsatisfactory, whereas the rest were, on average, between good and satisfactory. This may be because the surpluses law of Témez is asymptotic to the one proposed by Thornthwaite-Mather for the highest precipitation values, but it differs on the lowest side of the curve without needing both PET and soil moisture deficit to be used up completely.*

*With regard to the average of the models' results in each basin (Fig. 6), the influence of the aridity index is highlighted, as mentioned before,* which indicates *that the climatic characteristics of the watershed are the most important issue in a model's performance. In all basins, from the HOY catchment to the driest catchment, the NSE values (Fig. 6, upper left) were considered unsatisfactory on average, taking into account all the models studied. The same occurred in relation to RSR (Fig. 6, bottom left), with values below 0.70 in the HOY catchment and drier ones. Due to the better results obtained for the PBIAS (Fig. 6, upper right) in all catchments except for the TAM catchment, the Bressiani criteria showed unsatisfactory results in the four basins that were drier than the JUB catchment and in the HOY catchment (Fig. 6, bottom right).*

*Table 6 summarizes the classification sum according to the Bressiani criteria for all studied catchments and models in both the calibration (1980-1995) and validation (1995-2010) periods. The GR2M gave the best fit of the catchments, with nearly 90% in the calibration period and 50% in the validation period with values over 7, meaning they all had a very good fit. The disparities between calibration and validation may be caused because of the climatic differences between the calibration and validation periods, especially with flash flows occurring in many watersheds in the last fifteen years. The rest of the models achieved similar results in both the calibration and validation periods but lower than the GR2M model, though all semiarid catchments were unsatisfactory for the GR2M and for the rest of the models studied.* These watersheds produce frequent run-off peaks, and, although they have deep soil, low infiltration is produced, and therefore, models do not achieve good results. The *ABCD model had very similar results in five of the 16 catchments studied. The Témez and Thornthwaite-Mather models were the best in four and three catchments, respectively. However, the value obtained with the Témez model in the BOL catchment was just 4 in the validation period. The AWBM model was the best in only one catchment but showed high values (above 7) in some humid and sub-humid regions, corroborating the better performance of nearly all models in wetter conditions. In general, the models that showed the best results, on average, in all catchments were, first, the GR2M and then the ABCD and Thornthwaite-Mather, with values around 7.7/4.8 (calibration/validation) for the GR2M model and around 4.5 (calibration/validation) for the ABCD and Thornthwaite-Mather models. The AWBM and Témez showed the worst results, with 3.6 and 3.0, respectively, which is almost unsatisfactory. Moreover, the Témez model showed the highest coefficient of variation (129.22%), far from the rest of the models, the average value of which was around 80%. As with previous comparison methods, the best results were obtained in more humid basins, and the drier regions showed more unsatisfactory results. An ensemble model was performed using the average of all models (Table 6), showing that the results again depended on the aridity of the region, and there was not a great difference between the single models assessed and the ensemble model. The humid and sub-humid catchments of the ensemble model could be classified as very good and good except for the PUE and COT catchments. However, the results of the dry sub-humid catchments were less homogenous, especially in the less arid regions, with classifications from good to unsatisfactory, as in the HOY catchment. The more arid regions (from BOL to RVA) did not show satisfactory results in an ensemble average model. All the analyses taken into account did not show correlation between catchment area and goodness-of-fit results,*

*so higher* model resolution does not seem to be an improvement for humid and sub-humid watersheds (El Nasr et al., 2005; Booij, 2005; Refsgaard and Knudsen, 1996).

*The main aim of hydrological balance models is to assess inflows in a water resource system, and it is essential for appropriate analysis of its management. Therefore, in addition to the assessment performed with the Bressiani classification (Table 6), we should analyse the differences between total observed and simulated run-off volumes to validate or discard a model. Table 7 shows the REV results of these comparisons for the catchments studied and the models considered. According to this criterion, the Guo-5p shows the best results, on average, giving the best fitted model in six of the catchments studied. Furthermore, the Guo-5p model is the only water balance model that gave REV values below 15% in semiarid and dry sub-humid regions, despite the unsatisfactory results obtained in accordance with the Bressiani criteria (Table 6). The other models gave an average REV above 50% in these catchments. With regard to humid and sub-humid catchments, the lowest REV values accord with the Bressiani classification in most cases, which allows for selecting the best model in each catchment (Table 8).*

*Table 8 shows the proposed model for each catchment, taking into account the proposed set of criteria: R, the Bressiani classification and REV. R does not differ greatly between the various regions, altitudes or areas, although there is a slight descending trend (from 0.91 to 0.80) when we move to less humid catchments. HOY is the only dry sub-humid catchment where the correlation coefficient was less than 0.60, besides obtaining a sum of only 5 according to the Bressiani criteria, which is far from the average of 7.5 of its aridity group. This may be because, despite being a small basin (66.15 km²), it is the highest catchment of those studied. Nonetheless, the Thornthwaite-Mather model was classified as good in this catchment, and REV was below 2%. The GR2M, the Témez and the Guo-5p models were considered the models that best fit in humid regions, though in these regions, almost all models showed good results, with higher percentages in the GR2M model, as discussed before. The Thornthwaite-Mather gave the best results in humid sub-humid regions, and the Témez gave the worst fit, giving a classification of "unsatisfactory" in all catchments. However, despite the "very good" classification, REV percentages in the less humid catchments (ZUM and JUB) were around 20%. The driest and semiarid catchments did not have "satisfactory" classifications with any of the studied models, but REV was lower than 10% when the Guo-5p model was used. In contrast, the Témez model's "satisfactory" classification in the BOL catchment had an overestimate of nearly 30%. No trend was found for catchment area or altitude and the models used.*

*Since the ultimate goal in this study is to use model predictions for water management, flow duration curves (FDCs) for the observed and simulated streamflows for the whole studied period (1980-2010) have been made (Fig. 7) in order to assess quality metrics related to different segments of streamflow series. As has been analysed before, the more humid the watershed, the best performance obtained (almost) regardless of the model used. All the watersheds had a similar pattern, in general terms, except the TAM, CUE and RVA, which were the driest of those studied. The graphics confirm the previous values of Bressiani classification, only showing good performance in the highest volumes. Concerning the humid and sub-humid regions, almost all models had a good performance in probabilities of exceedance lower than 80%, i.e., in high and medium volumes, but both curves were separated the lower the streamflow was. Table 8 quantifies the variabilities identified by FDCs, using the signatures for high, low and midsegment volumes, described before. Deviations in humid and sub-humid watersheds ranged from 0% to 30% in both high volumes and midsegment slopes. However, values in low volumes for humid watersheds showed an average of 41% but reached values higher than 100% in PUE, TRE and PRI*

*watersheds (GR2M and Témez models). The best performance was obtained in the AND watershed with values below 25% for all the signatures. Nonetheless, low volumes in most humid watersheds accounted for less than 5% of the total streamflow in the watersheds in the studied period, so their importance is relative for water resources management. The differences in low volumes increased the less humid the watershed was. While high volumes' differences stood at around 15-30%, signatures in low volumes reached to negative 200-400%, overestimating streamflows in spring and summer seasons. As with humid regions, volumes with probability of exceedance above 80% accounted for less than 8% of the total volume in the studied period. On the contrary, semiarid regions showed good performance in high volumes, but both midsegment slopes and low volumes had differences above 100% in most cases, and high volumes in these watersheds accounted for only 25% of the total streamflows in the studied period.*

*NEW REFERENCES*

Abu El-Nasr, A., Arnold, J. G., Feyen, J. &Berlamont, J. Modelling the hydrology of a catchment using a distributed and a semi-distributed model. Hydrol.Processes 19, 573–587, 2005.

Bai, P., Liu, X., Liang, K., Liu, Ch., Comparison of performance of twelve monthly balance models in different climatic catchments of China, J. Hydrol., 529, 1030-1040, 2015.

Booij, M.J. Impact of climate change on river flooding assessed with different spatial model resolutions. J. Hydrol., 303, 1-4, 176-198.

Beven,K.Changing ideas in hydrology—the case of physically-based models.J. Hydrol.,105, 1-2, 157-172, 1989.

*Clark, M. P., Slater, A. G., Rupp, D. E., Woods, R. A., Vrugt, J. A., Gupta, H. V., Wagener, T., Hay, L. E. Framework for Understanding Structural Errors (FUSE): A modular framework to diagnose differences between hydrological models. Water Resour. Res., 44, W00B02, doi:10.1029/2007wr006735, 2008*

Coron, L., Andréassian, V., Perrin, C., Lerat, J., Vaze, J., Bourqui, M., and Hendrickx, F. Crash testing hydrological models in contrasted climate conditions: An experiment on 216 Australian catchments., Water Resour. Res., 48, W05552, doi:10.1029/2011wr011721, 2012.

Jakeman, A. J. and Hornberger, G. M.: How much complexity is warranted in a rainfall-runoff model?, Water Resour. Res., 29, 2637–2649, 1993

Michaud, J., Sorooshian, S., Comparison of simple versus complex distributed runoff models on a midsized semiarid watershed. Water Resour.Res., 30, 3, 593-605, 1994.

Perrin, C., Michel, C., Andréassian, V.Does a large number of parameters enhance model performance? Comparative assessment of common catchment model structures on 429 catchments. J.Hydrol., 242, 3, 275-301,2001.

Refsgaard, J.C. and Knudsen, J. (1996). Operational validation and intercomparison of different types of hydrological models. Water Resour. Res., 32: doi: 10.1029/96WR00896. issn: 0043-1397.

vanEsse, W. R., Perrin, C., Booij, M. J., Augustijn, D. C. M., Fenicia, F., Kavetski, D., Lobligeois, F. The influence of conceptual model structure on model performance: a comparative study for 237 French catchments.Hydrol.EarthSyst. Sci., 17, 4227-4239, https://doi.org/10.5194/hess-17-4227-2013, 2013

WMO, Intercomparison of conceptual models used in operational hydrological forecasting. Secretariat of the WMO, 1975.

Yew Gan, T., Dlamini, E.M., Biftu, G.F.,.Effects of model complexity and structure, data quality, and objective functions on hydrologic modeling. J. Hydrol., 192, 1, 81-103, 1997

**9) Page 6, line 25: To which period (calibration or verification) these results belong? Check also elsewhere. Could you present the calibration and validation period results separately? Is the model ranking similar between two periods?**

That sentence and all the tables included in the original version refer to the validation period. The models rank, in most cases, similar in both periods. In the final version, we will add to Tables 5 and 6 both the calibration and validation values, as follows:

*Table 5.* *Correlation coefficient for observed-simulated streamflows (calibration: 1980-1995/validation: 1995-2010). Best fit (highest value for each watershed) in validation period in bold.*

| Catchment | ABCD | AWBM | GR2M | GUO-5P | Témez | THOR-MATH | Average |
|---|---|---|---|---|---|---|---|
| PUE | 0.79/0.79 | 0.82/0.84 | **0.86/0.91** | 0.81/0.83 | 0.83/0.84 | 0.77/0.77 | 0.97/0.86 |
| AND | 0.94/0.90 | 0.94/0.90 | **0.95/0.91** | 0.93/0.89 | **0.95**/0.90 | 0.94/0.90 | 0.94/0.91 |
| BEG | 0.92/0.92 | 0.92/0.92 | **0.93/0.93** | 0.92/0.91 | 0.92/0.92 | 0.92/0.92 | 0.92/0.92 |
| LEM | 0.81/0.89 | 0.80/0.89 | **0.83/0.93** | 0.81/0.88 | 0.82/0.90 | 0.82/0.88 | 0.82/0.90 |
| TRE | 0.90/0.90 | 0.90/0.90 | **0.92/0.92** | 0.90/0.90 | 0.90/0.89 | 0.90/0.89 | 0.90/0.90 |
| COT | 0.87/0.85 | 0.85/0.86 | **0.91/0.91** | 0.87/0.85 | 0.90/0.89 | 0.87/0.85 | 0.88/0.88 |
| PRI | 0.86/0.87 | 0.85/0.88 | **0.88/0.89** | 0.86/0.88 | 0.88/**0.89** | 0.86/0.87 | 0.86/0.88 |
| GAR | 0.90/0.88 | 0.90/0.90 | **0.93**/0.90 | 0.83/0.92 | 0.90/**0.92** | 0.91/0.90 | 0.90/0.90 |
| HOY | 0.79/0.66 | 0.79/0.66 | **0.83/0.67** | 0.81/0.67 | 0.79/0.66 | 0.68/0.58 | 0.78/0.65 |
| SEG | 0.91/0.83 | 0.91/0.83 | **0.95/0.84** | 0.84/0.84 | 0.93/0.72 | 0.93/**0.84** | 0.91/0.82 |
| ZUM | 0.82/0.77 | 0.89/0.77 | **0.92/0.82** | 0.88/0.60 | 0.83/0.64 | 0.82/0.66 | 0.86/0.71 |
| JUB | 0.81/0.78 | 0.88/0.83 | **0.88/0.85** | 0.86/0.82 | 0.85/0.79 | 0.89/0.83 | 0.86/0.82 |
| BOL | 0.81/0.76 | 0.78/0.65 | **0.84**/0.74 | **0.84/0.78** | 0.77/0.73 | 0.79/0.72 | 0.81/0.73 |
| TAM | 0.75/0.89 | **0.76**/0.92 | **0.76/0.90** | 0.70/0.89 | 0.76/0.89 | 0.43/0.59 | 0.69/0.85 |
| CUE | 0.52/0.71 | 0.70/0.87 | **0.70/0.90** | 0.68/0.89 | 0.67/0.89 | 0.69/0.87 | 0.66/0.86 |
| RVA | 0.68/0.61 | 0.67/0.75 | **0.76/0.85** | 0.70/0.80 | 0.66/0.59 | 0.49/0.52 | 0.66/0.69 |
| Best fit (Number of times) | 0/0 | 1/0 | 16/14 | 1/1 | 1/2 | 0/1 | |

*Table 6.* *Bressiani classification values (calibration: 1980-1995/validation: 1995-2010). Best fit (highest value for each watershed) in bold.*

| Catchment | ABCD | AWBM | GR2M | GUO-5P | Témez | THOR-MATH | Average | Ensemble Classification |
|---|---|---|---|---|---|---|---|---|
| PUE | 5/3 | 0/0 | **7/8** | 3/7 | 0/0 | 3/5 | 3.0/3.8 | S |

| | | | | | | | | |
|---|---|---|---|---|---|---|---|---|
| AND | 8/**9** | 8/7 | **9/9** | **9/9** | 8/7 | **9**/8 | 8.5/8.2 | V |
| BEG | **9/9** | 8/8 | **9/9** | **9/9** | 8/8 | 8/8 | 8.5/8.5 | V |
| LEM | 5/4 | 0/7 | **7**/0 | 5/3 | 3/**9** | 5/7 | 4.2/5.0 | G |
| TRE | **9/9** | 7/7 | **9/9** | **9**/8 | 8/**9** | 7/8 | 8.2/8.3 | V |
| COT | 7/4 | 0/0 | **9**/5 | 6/7 | 0/2 | 6/**9** | 4.7/4.5 | S |
| PRI | 7/7 | 7/7 | **9/9** | 7/7 | 6/8 | 7/7 | 7.2/7.5 | G |
| GAR | 7/4 | 6/**9** | **9**/7 | 0/**9** | **9**/0 | 6/**9** | 6.2/6.3 | G |
| HOY | 0/4 | 0/0 | **7**/0 | 4/0 | 0/0 | 0/**5** | 1.8/1.5 | U |
| SEG | 3/**9** | **9**/5 | **9**/3 | **9**/4 | 8/3 | **9**/8 | 7.8/5.3 | G |
| ZUM | 3/**7** | **9**/5 | **9**/7 | **9**/5 | 3/0 | 4/5 | 6.2/4.8 | G |
| JUB | 0/5 | **9**/3 | **9**/4 | 7/3 | 6/0 | **9**/7 | 6.7/3.7 | S |
| BOL | 5/3 | 0/0 | **7**/0 | 5/0 | 3/**4** | 0/3 | 3.3/1.7 | U |
| TAM | 0/0 | 6/0 | 0/**6** | **9**/3 | 4/0 | 0/0 | 3.2/1.5 | U |
| CUE | 0/0 | 0/0 | 0/0 | 0/0 | 0/0 | 0/0 | 0.0/0.0 | U |
| RVA | 0/0 | 0/0 | **5**/0 | 0/0 | 0/0 | 0/0 | 0.0/0.0 | U |
| Average | 4.2/4.8 | 4.3/3.63 | 7.7/4.8 | 5.7/4.6 | 4.1/3.0 | 4.6/4.6 | | |
| Best fit (Number of times) | 3/5 | 3/1 | 14/7 | 6/3 | 1/3 | 3/5 | | |

**Table 7.** REV (%) values in period 1980-2010. Best fit (lowest absolute value for each watershed) in bold.

| Catchment | ABCD | AWBM | GR2M | GUO-5P | Témez | THOR-MATH |
|---|---|---|---|---|---|---|
| PUE | -2.95 | -29.46 | 2.67 | -2.57 | -29.98 | **0.60** |
| AND | -10.09 | -15.46 | **1.05** | -3.68 | -15.78 | -13.86 |
| BEG | -5.02 | -10.06 | 21.61 | **0.66** | -8.70 | -9.81 |
| LEM | 16.24 | **-4.49** | -16.22 | -23.47 | 4.62 | 23.55 |
| TRE | 8.20 | -7.55 | 37.68 | 14.06 | **0.74** | -2.86 |
| COT | **-1.20** | -41.41 | 2.19 | -4.34 | -36.84 | 3.43 |
| PRI | **-0.80** | -8.24 | 15.48 | 11.59 | -10.28 | -5.27 |
| GAR | 63.44 | 45.58 | 22.27 | **3.43** | 54.02 | 52.81 |
| HOY | 11.20 | -35.00 | 31.60 | 17.78 | -32.90 | **-1.75** |

| | | | | | | |
|---|---|---|---|---|---|---|
| SEG | 20.24 | 5.59 | **-0.75** | -4.07 | 5.43 | 11.21 |
| ZUM | 16.25 | **-3.87** | 16.81 | 6.72 | 26.54 | 7.25 |
| JUB | -18.60 | **-17.92** | 88.28 | 52.69 | -26.40 | -21.58 |
| BOL | 37.20 | 19.25 | -30.62 | **12.39** | 27.70 | 20.38 |
| TAM | 113.54 | 64.59 | -37.41 | **-3.61** | 97.40 | -18.45 |
| CUE | 32.61 | 27.92 | 48.22 | **13.61** | 38.61 | 47.20 |
| RVA | -26.95 | -34.30 | 24.74 | **1.04** | -77.66 | -78.97 |
| Average (absolute value) | 24.03 | 23.17 | 24.85 | 10.98 | 30.85 | 19.94 |
| Best fit (Number of times) | 2 | 3 | 2 | 6 | 1 | 2 |

**10) Page 8, line 1: What do you mean by "irregular"?**

We mean that in those regions, results vary significantly and do not show a clear tendency for either aridity or physical characteristics. This expression, as well as the Discussion section, will be modified for better comprehension and a greater research focus of the paper.

**11) I find caption of nearly all figures not very informative. For example in Figure 4 – is it for the calibration or verification period? Unit of flow should follow the SI system. Figure 5 and 6 – again to which period, does these results correspond and to which river basins? Figure 7 - please provide indication to calibration and validation periods.**

The captions in Figures 4-6 will be modified in order to specify the period included, as follows:

**Fig. 4.** Scatter plots (observed and simulated streamflows in $10^6 m^3 month^{-1}$) for best model fit according to Pearson's correlation coefficient (R) in the validation period (1995-2010). The dashed line is the estimated regression line, and the solid line is the perfect fit.

**Fig. 5.** Model average comparison in validation period (1995-2010).

**Fig. 6.** Comparison of catchment averages of goodness-of-fit tests in validation period (1995-2010).

The captions in Tables 5-7 will be also modified (as shown above) as follows:

**Table 5.** *Correlation coefficient for observed-simulated streamflows (calibration: 1980-1995/validation: 1995-2010). Best fit (highest value for each watershed) in validation period in bold.*

**Table 6.** *Bressiani classification values (calibration: 1980-1995/validation: 1995-2010). Best fit (highest value for each watershed) in bold.*

**Table 7.** *REV (%) values in period 1980-2010. Best fit (lowest absolute value for each watershed) in bold.*

Figure 7 will be replaced by the flow duration curves, as follows:

[Figure]

**Fig. 7.** Flow duration curves for observed and selected models values in the study period (1980–2010).

**12) Figure 2: What information is gained by having this figure in the paper - once we have already Fig 1. The quality of Figures 2 and 7 is also very poor. Hard to visualize them and also distinct features could not be identified as presented.**

Figure 2 will be deleted, and Figure 7 will be replaced by the flow duration curves, as shown above.

**13) Overall I find there are lots of tables with lots of information inside - that are not thoroughly discussed in the text. Try reducing them to only couple of informative tables and discuss them in detail. Tables like 2 and 3 could be easily dropped out do not add much to the overall message of this paper.**

Tables 2 and 3 will be deleted in the final version of the paper. Table 2 will be replaced by only the equations of the goodness-of-fit tests, and the content of Table 3 will be briefly explained in the text.

---

## Author Comment (AC2) · 2 Dec 2017

We appreciate the editor giving us the opportunity to improve the paper during the review process. Following the editor's suggestions, we have replied to each of the reviewers' comments. The answers are detailed in the attached file (hess_2017_424_supplement.pdf) as supplementary material. Based on these answers, if the editor considers it appropriate, we will prepare the revised version of the manuscript.

Please also note the supplement to this comment: https://www.hydrol-earth-syst-sci-discuss.net/hess-2017-424/hess-2017-424-AC2-supplement.pdf